# Avoiding Leakage Poisoning: Concept Interventions Under Distribution Shifts

**Mateo Espinosa Zarlenga** [1]  **Gabriele Dominici** [2]  **Pietro Barbiero** [3]  **Zohreh Shams** [1 4]  **Mateja Jamnik** [1]

## Abstract

In this paper, we investigate how concept-based models (CMs) respond to out-of-distribution (OOD) inputs. CMs are interpretable neural architectures that first predict a set of high-level *concepts* (e.g., stripes, black) and then predict a task label from those concepts. In particular, we study the impact of *concept interventions* (i.e., operations where a human expert corrects a CM's mispredicted concepts at test time) on CMs' task predictions when inputs are OOD. Our analysis reveals a weakness in current state-of-the-art CMs, which we term *leakage poisoning*, that prevents them from properly improving their accuracy when intervened on for OOD inputs. To address this, we introduce MixCEM, a new CM that learns to dynamically exploit leaked information missing from its concepts only when this information is in-distribution. Our results across tasks with and without complete sets of concept annotations demonstrate that MixCEMs outperform strong baselines by significantly improving their accuracy for both in-distribution and OOD samples in the presence and absence of concept interventions.

## 1. Introduction

Recent years have seen a surge of interpretable models whose performance is comparable to that of powerful black-box models such as Deep Neural Networks (DNNs) (Alvarez-Melis & Jaakkola, 2018; Chen et al., 2019; Yuksekgonul et al., 2023). Amongst these, concept-based models (CMs) (Chen et al., 2020; Espinosa Zarlenga et al., 2022), and in particular Concept Bottleneck Models (CBMs) (Koh et al., 2020), have paved the way for designing expressive yet interpretable models. CBMs and their variants predict downstream task labels by exploiting high-level units of information known as "*concepts*" (e.g., "stripes", "white", and "black" when predicting "zebra"). They achieve this through a two-step process: first, they predict concepts from the inputs, forming a *bottleneck*, and then they use these concept predictions to determine the task label. This design enables CBMs to provide human-like explanations for their predictions by grounding them in interpretable concept representations. More importantly, these models enable *concept interventions* (Koh et al., 2020; Chauhan et al., 2022; Sheth et al., 2022; Shin et al., 2023), where an expert interacting with the model at test time can correct mispredicted concepts, leading to significant improvements in accuracy once the CBM updates its prediction considering such feedback (Figure 1, left).

Recent works have significantly advanced the performance and usability of CMs within challenging in-distribution (ID) test sets by overcoming the *incompleteness gap* – the fact that training concept annotations may be insufficient for accurately predicting the downstream task. By exploiting bypass mechanisms, such as dynamic concept embeddings (Espinosa Zarlenga et al., 2022; Kim et al., 2023; Xu et al., 2024) or residual connections (Mahinpei et al., 2021; Havasi et al., 2022; Yuksekgonul et al., 2023), state-of-the-art CMs enable information to "*leak*" directly from the features to the task predictions, bypassing the concept bottleneck and significantly increasing the model's task accuracy even when the set of training concept annotations is *incomplete*.

In this work, we argue that, although useful, blindly incorporating these bypasses can severely affect how CMs behave for out-of-distribution (OOD) samples. Specifically, we suggest that such bypasses can themselves become out-of-distribution for OOD samples, resulting in the "*poisoning*" of the model's predictions and in concept interventions failing to achieve the intended accuracy improvements (Figure 1, right). Given how interventions can aid a CM in adjusting to real-world OOD shifts (e.g., an expert can help a CM process a noisy yet still-interpretable X-ray scan by intervening on some concepts), such *leakage poisoning* casts serious doubts on the real usability of existing CMs.

To address these limitations, we propose the *Mixture of Concept Embeddings Model (MixCEM)*, a concept-based interpretable model with high generalisation and receptiveness to interventions across data distributions. MixCEMs

---

[1]University of Cambridge [2]Università della Svizzera Italiana [3]IBM Research [4]Leap Laboratories Inc.. Correspondence to: Mateo Espinosa Zarlenga <me466@cam.ac.uk>.

*Proceedings of the $42^{nd}$ International Conference on Machine Learning*, Vancouver, Canada. PMLR 267, 2025. Copyright 2025 by the author(s).

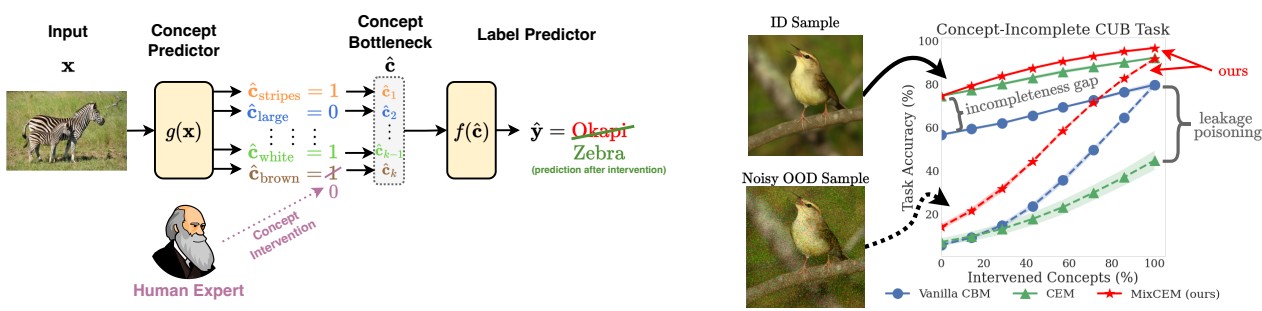

Figure 1. **(Left)** A concept intervention on a CBM triggering a prediction update. **(Right)** Task accuracy as concepts are intervened on in a concept-incomplete task. When intervening on ID samples (solid), bypass-enabling models (e.g., CEMs) overcome the "incompleteness gap." However, for OOD samples (dashed), the same models underperform due to "leakage poisoning." MixCEM overcomes the incompleteness gap and leakage poisoning, maintaining high accuracy in both setups.

achieve this by learning, for each concept, an embedding formed by mixing a *global*, sample-agnostic embedding and a *contextual*, residual embedding. By introducing a confidence-based gating mechanism to control the residual embedding's contribution, MixCEMs learn to decide when the residual information may be detrimental and, therefore, should be dropped. This allows MixCEMs to exploit the residual component when a bypass is needed (e.g., concept-incomplete setups) while dropping it for OOD samples, leading to impactful interventions across data distributions.

**Summary of Contributions** Our main contributions are: (1) we provide the first study, to the best of our knowledge, of how concept interventions fare when there are distribution shifts. Our experiments suggest that all bypass-based approaches do not necessarily or significantly improve their OOD task accuracy when intervened on; (2) we introduce the notion of *leakage poisoning*, a previously unknown consequence of *information leakage* (Mahinpei et al., 2021). Then, we argue that this poisoning is an important design consideration given the existence of a trade-off between avoiding leakage poisoning and achieving high accuracies; and (3) we propose MixCEM, a CM that avoids leakage poisoning[1]. We show that MixCEMs maintain high task and concept accuracies while significantly improving their performance when intervened on, both for OOD and ID samples and even in concept-incomplete training sets.

## 2. Background, Notation, and Related Work

**Concept-based Explainable AI (C-XAI)** Concept-based XAI methods explain a black-box model's predictions via high-level units of information, or *concepts*, that experts would use to explain the same task (Bau et al., 2017). Such concepts, which can be provided as training labels (Chen et al., 2020; Kazhdan et al., 2020; Rigotti et al., 2021;

Crabbé & van der Schaar, 2022; Sheth & Ebrahimi Kahou, 2024) or can be discovered (Alvarez-Melis & Jaakkola, 2018; Ghorbani et al., 2019; Yeh et al., 2020; Magister et al., 2022; Espinosa Zarlenga et al., 2023b; Yang et al., 2023; Oikarinen et al., 2023), enable these methods to circumvent the unreliability (Kindermans et al., 2017; Adebayo et al., 2018) and lack of semantic alignment (Kim et al., 2018) of traditional XAI *feature importance* approaches (Ribeiro et al., 2016; Erhan et al., 2009; Lundberg & Lee, 2017).

Within C-XAI, Concept Bottleneck Models (CBMs) (Koh et al., 2020) provide a powerful framework for designing concept-based interpretable DNNs. A CBM $\left(g, f, \{s_i\}_{i=1}^k\right)$ is a composition of two functions $(g, f)$ supported by *scoring* functions $\{s_i\}_{i=1}^k$, all usually parameterised as DNNs. The *concept encoder* $g : \mathbb{R}^n \to \mathcal{C}^k$ maps an input $\mathbf{x} \in \mathbb{R}^n$ to a "bottleneck" $\hat{\mathbf{c}} = g(\mathbf{x}) \in \mathbb{R}^{k \times m}$ of $k$ concepts in *concept space* $\mathcal{C} \subseteq \mathbb{R}^m$. Here, the $i$-th output of $g$, $\hat{\mathbf{c}}_i = g(\mathbf{x})_i$, is designed such that the score $\hat{p}_i := s_i(\hat{\mathbf{c}}_i)$ is maximised when the $i$-th concept is "*active*", and minimised otherwise. The *label predictor* $f : \mathcal{C}^k \to \mathbb{R}^L$ maps the bottleneck $\hat{\mathbf{c}}$ to a distribution over $L$ task labels $\hat{\mathbf{y}} \in \mathbb{R}^L$. Together, these functions predict a label $\hat{y} = f(g(\mathbf{x}))$ for a sample $\mathbf{x}$ that can be explained via the concept scores $s(\hat{\mathbf{c}}) := [s_1(\hat{\mathbf{c}}_1), \cdots, s_k(\hat{\mathbf{c}}_k)]^T$. When $m = 1$ and $s_i(\hat{\mathbf{c}}_i) = \hat{\mathbf{c}}_i$, we call this a *Vanilla CBM* (i.e., Koh et al.'s formulation). Vanilla CBMs can be trained *jointly* (optimising $f$ and $g$ together), *sequentially* (training $g$ first and then $f$ using $g$'s outputs), or *independently* (training $g$ and $f$ using ground-truth features and concepts as inputs).

**CBM Extensions** Recent works have addressed several limitations of Vanilla CBMs: (1) *Concept Embedding Models* (CEMs) (Espinosa Zarlenga et al., 2022) and Intervention-aware CEMs (IntCEMs) (Espinosa Zarlenga et al., 2023a) overcome the aforementioned *incompleteness gap* (Yeh et al., 2020), (2) *Probabilistic CBMs* (ProbCBMs) (Kim et al., 2023) and *Energy-based CBMs* (ECBMs) (Xu et al., 2024) enable better uncertainty and

---

[1]Our code and experiment configs can be found at https://github.com/mateoespinosa/cem

conditional probability estimations, (3) *Post-hoc CBMs* (P-CBMs) (Yuksekgonul et al., 2023) allow for effective fine-tuning of models into CBMs, and (4) *Label-free CBMs* (Oikarinen et al., 2023; Yang et al., 2023) exploit vision-language models to extract concept annotations.

**Concept Interventions**   CBM-based models enable *concept interventions*, where a human-in-the-loop can correct mispredicted concepts at test time, potentially triggering a change in task prediction. Formally, an intervention on concept $c_i$ fixes $s_i(\hat{\mathbf{c}}_i)$ to its maximum if the expert determines $c_i$ is active or to its minimum otherwise. Previous works have shown that CBM-based models can significantly increase their task accuracy when the corrected concepts are carefully selected via an *intervention policy* (Shin et al., 2023; Chauhan et al., 2022), or even when they are randomly selected (Koh et al., 2020; Espinosa Zarlenga et al., 2022; Xu et al., 2024). Recent works have improved the effect of interventions by incorporating intervention-aware losses (Espinosa Zarlenga et al., 2023a), intervention memories (Steinmann et al., 2023), or cross-concept relationships (Havasi et al., 2022; Vandenhirtz et al., 2024).

**OOD Detection**   This paper studies concept interventions when OOD shifts occur. Hence, our work is related to research in OOD generalisation (Sagawa et al., 2019), open set recognition (Scheirer et al., 2012), and anomaly (Zhou & Paffenroth, 2017; Schlegl et al., 2017) and distribution shift (Rabanser et al., 2019) detection. Within concept-based XAI, concepts have been used to explain distribution shifts (Wijaya et al., 2021; Sevyeri et al., 2023; Choi et al., 2023; Dreyer et al., 2024), while OOD detectors have been used to detect unwanted leakage in concept representations (Marconato et al., 2022). Rather than explaining shifts or detecting leakage, our work focuses on understanding concept interventions when OOD shifts occur.

## 3. Conflicting Objectives in CBMs

CBMs have been traditionally designed with three core objectives in mind: (1) *task fidelity* (the model should accurately predict its task), (2) *concept fidelity* (the model's explanations should be accurate), and (3) *intervenability* (Marcinkevičs et al., 2024) (task fidelity should improve when a model is intervened on). These three properties capture the fact that model *trustworthiness*, under reasonable definitions of the term (Shen, 2022), cannot rely solely on concept and task fidelity without incorporating intervenability. It is important, however, to place CBMs within the context of real-world datasets, where labelling limitations and distribution shifts are commonplace. Considering this, we argue that CBMs ought to have two additional properties:

1. **Completeness-Agnosticism (CA)**: Task fidelity should be independent of the set of training concepts.

That is, if (i) $(f^{(\mathcal{P}_{\mathrm{tr}})}, g^{(\mathcal{P}_{\mathrm{tr}})}, \{s_i^{(\mathcal{P}_{\mathrm{tr}})}\}_{i=1}^k)$ is a CBM learnt from any training distribution $\mathcal{P}_{\mathrm{tr}}$, and (ii) $\mathcal{P}_*(X, C_*, Y)$ is a *concept-complete* distribution (i.e., $I(X; Y) \leq I(C_*; Y)$, where $I(\cdot)$ is the mutual information), then $f^{(\mathcal{P}_*)}(g^{(\mathcal{P}_*)}(\mathbf{x})) \approx f^{(\mathcal{P}_{\mathrm{tr}})}(g^{(\mathcal{P}_{\mathrm{tr}})}(\mathbf{x}))$. This condition implies that, regardless of the set of concepts used to train the CBM, its downstream task accuracy should remain relatively unchanged. Therefore, a model achieves CA if it can perform equally well (in terms of task fidelity) regardless of the span of its training concept set.

2. **Bounded Intervenability (BI)**: For *any* test distribution $\mathcal{P}_{\mathrm{te}}$ and any concept subset $S \subseteq \{1, \cdots, k\}$, when concepts in $S$ are intervened on with values $\mathbf{c}_S$, a CBM's task accuracy should be at least as high as the accuracy of a Bayes Classifier in the real data distribution $\mathcal{P}_{\mathrm{d}}$ given $\mathbf{c}_S$:

$$\mathbb{E}_{(\mathbf{x},\mathbf{c},y) \sim \mathcal{P}_{\mathrm{te}}}\Big[\mathcal{I}(\mathbf{x}, S, \mathbf{c}_S)_y\Big] \geq \mathbb{E}_{(\mathbf{x},\mathbf{c},y) \sim \mathcal{P}_{\mathrm{d}}}\Big[\mathbb{P}(y \mid \mathbf{c}_S)\Big]$$

where $\mathcal{I}(\mathbf{x}, S, \mathbf{c}_S)$ is the CBM's task prediction after intervening on concepts $S \subseteq \{1, \cdots, k\}$ using values $\mathbf{c}_S$. Notice that the magnitude of $\mathbb{E}_{(\mathbf{x},\mathbf{c},y) \sim \mathcal{P}_{\mathrm{te}}}[f(g(\mathbf{x}))_y]$ is equivalent to the expected *accuracy* of the CBM $(g, f, \{s_i\}_{i=1}^k)$ on a test set sampled from $\mathcal{P}_{\mathrm{te}}$. Intuitively, BI says that, when we are given the ground-truth labels for concepts $S$, we should strive to perform at least as well as a model that **only** has access to the labels for concepts in $S$.

Completeness-agnosticism yields task-accurate CBMs even when their training sets lack all task-relevant concepts, a common scenario considering the difficulty of labelling concepts in both supervised (Collins et al., 2023) and unsupervised (Oikarinen et al., 2023) settings. In contrast, bounded intervenability ensures that a CBM will properly incorporate interventions, even for out-of-distribution inputs, providing a sensible and computable lower bound for the post-intervention accuracy.

**Leakage Poisoning**   State-of-the-art CBMs have embedded incentives within their loss functions and architectures for each core objective. More recently, new architectures have incorporated notions similar to that of CA in their design (e.g., leakage bypasses). The notion of bounded intervenability, however, remains overlooked in the design and evaluation of CBMs. In this section, we argue that this disregard for BI has led to a serious limitation of current CA-supporting models to remain unnoticed.

In Figure 1 (right), we observe that completeness-agnostic methods like CEMs indeed overcome the "completeness gap", attaining high task accuracies compared to non-completeness-aware approaches (e.g., Vanilla CBMs) on

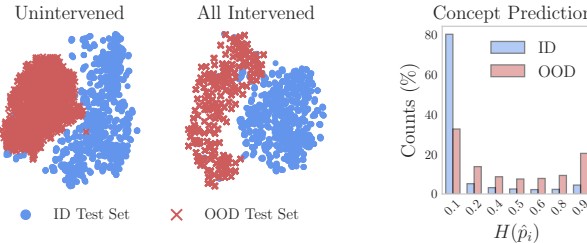

*Figure 2.* CEM concept bottlenecks and predicted concept entropies for ID and noisy (OOD) test CUB samples. **(Left and centre)** t-SNE projections of CEM's bottlenecks before and after all concepts are intervened on. **(Right)** Distribution of predicted concept entropies for all concepts.

concept-incomplete tasks. However, we also observe that these approaches struggle to properly incorporate interventions for OOD inputs, even when the shift is subtle random noise. Surprisingly, we see that when all concepts are intervened on, CEM's accuracy is significantly worse for OOD samples than for ID samples, something not observed for Vanilla CBMs. This suggests OOD shifts somehow affect a CEM's bottleneck even after intervening on all concepts.

To understand this, we emphasise that state-of-the-art CBMs achieve CA by enabling information about $y$ missing in the concepts $\mathbf{c}$ to leak to the downstream task predictor $f$. In practice, this is done using dynamic high-dimensional concept embedding representations (e.g., CEMs, ProbCBMs, and ECBMs) or residual side-channels that extend or update the bottleneck after each concept's representation has been constructed (e.g., Hybrid CBMs (Mahinpei et al., 2021), autoregressive CBMs (Havasi et al., 2022), and residual P-CBMs). Although useful, the concept bottlenecks yielded by such models form very distinct distributions for ID and OOD samples (Figure 2, left). More importantly, these distributions remain distinct even after intervening on all concepts (Figure 2, centre). This is because, when we enable information not in $\mathbf{c}$ to leak into the bottleneck $\hat{\mathbf{c}}$ using high-dimensional embeddings or residual pathways, this information persists even after an intervention is performed. Hence, when an input goes OOD, the leakage path may go OOD itself, becoming detrimental, or "*poisonous*", for the model's ability to intake interventions in general.

**The Intervenability-Incompleteness Trade-off** The observed existence of leakage poisoning suggests there is a trade-off between satisfying completeness-agnosticism (which requires one to bypass information directly from $\mathbf{x}$ to $\mathbf{y}$ even *after* an intervention is performed) and satisfying bounded intervenability (which requires interventions to lead to ID bottlenecks even for OOD samples). Yet, our study reveals two other critical observations that will form the basis of our solution to leakage poisoning: first, the intervention curves for Vanilla CBM converge to the same

points for both ID and OOD samples. This is because interventions on Vanilla CBMs result in global *constant* changes to their bottlenecks (e.g., setting $\hat{\mathbf{c}}_i := 1$ if $\mathbf{c}_i$ is "active"). Hence, the more one intervenes, the more the bottleneck will become in-distribution. Second, concept predictions are significantly more uncertain (i.e., have higher entropy) for OOD samples (Figure 2, right). Thus, the uncertainty in $\hat{\mathbf{p}}$ is a helpful indicator of a sample going OOD, a property also exploited in OOD detection (Hendrycks & Gimpel, 2016).

## 4. Mixture of Concept Embeddings Model

In this section, we build upon our observations above to propose the *Mixture of Concept Embeddings Model (MixCEM)*. MixCEM is a novel CM where interventions can dynamically leak information when the sample is ID, hence enabling completeness-agnosticism, while they are reduced to *global* constant changes to the bottleneck when the input is OOD, hence avoiding leakage poisoning (Figure 3).

**Overview** Given a training set $\mathcal{D} = \{(\mathbf{x}^{(j)}, \mathbf{c}^{(j)}, y^{(j)})\}_{j=1}^{N}$ with $k$ human-generated or label-free concept annotations, MixCEM learns $k$ pairs of $m$-dimensional *global embeddings* $\bar{C} = \{(\bar{\mathbf{c}}_i^{(+)}, \bar{\mathbf{c}}_i^{(-)})\}_{i=1}^{k}$ such that concept $c_i$ is represented by $\bar{\mathbf{c}}_i^{(+)}$ when it is "active" and $\bar{\mathbf{c}}_i^{(-)}$ otherwise. These embeddings will be used for concept prediction and for constructing an intervenable bottleneck $\hat{\mathbf{c}}$ from which we predict task labels. However, to achieve completeness-agnosticism, MixCEM will learn to adjust these embeddings to allow task-relevant information missing in the concept annotations $\mathbf{c}$ to leak when this information is beneficial.

**Residual Embeddings** Given $\mathbf{x} \in \mathbb{R}^n$, we introduce a leakage mechanism in our concept embeddings by using a latent code $\mathbf{h} \in \mathbb{R}^a$, generated from a backbone model $\psi(\mathbf{x})$ (e.g., a pre-trained ResNet (He et al., 2016)), to construct a pair of *residual* concept embeddings $\left(r_i^{(+)}(\mathbf{x}), r_i^{(-)}(\mathbf{x})\right)$ for each concept $c_i$. We learn these residuals using two linear functions $r_i^{(+/-)}(\mathbf{x}) := \mathbf{R}_i^{(+/-)} \cdot \psi(\mathbf{x}) + \mathbf{b}_i^{(+/-)}$ with learnable weights $\mathbf{R}_i^{(+/-)}$ and biases $\mathbf{b}_i^{(+/-)}$. These residuals will be used to update our global embeddings.

**Concept Likelihood** From the global and residual embeddings, we estimate the likelihood $\hat{p}_i = \mathbb{P}(c_i = 1 \mid \mathbf{x}, \bar{C})$ using a linear scoring function $s_i(\mathbf{x}) = \sigma\Big(\mathbf{v}_s \cdot [\bar{\mathbf{c}}_i^{(+)} + r_i^{(+)}(\mathbf{x}), \bar{\mathbf{c}}_i^{(-)} + r_i^{(-)}(\mathbf{x})]^T\Big)$ with weights $\mathbf{v}_s \in \mathbb{R}^{2m}$ shared across concepts. This enables task and concept feedback to influence how we learn our global and residual embeddings.

**Contextual Concept Embeddings** By mixing each concept's global and residual embeddings as we did for $\hat{p}_i$, we can construct *contextual embeddings* $\mathbf{c}_i^{(+)}, \mathbf{c}_i^{(-)} \in \mathbb{R}^m$ that encode both a concept's global and sample-specific information. However, we want these contextual embed-

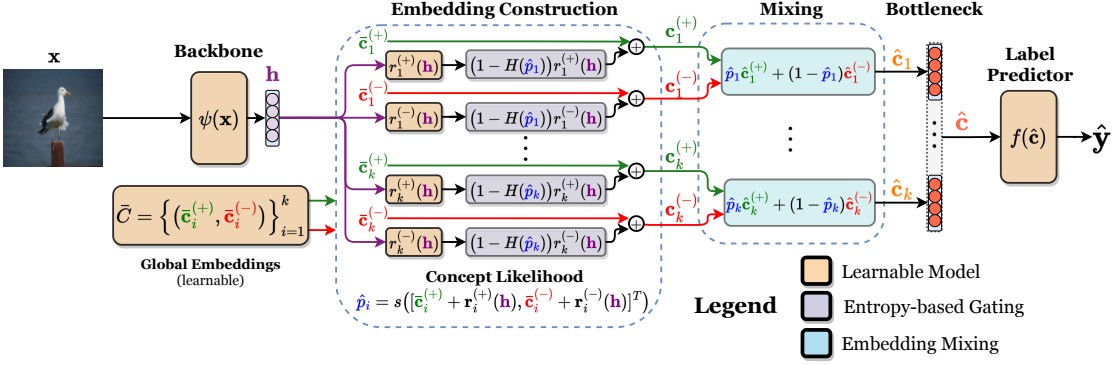

*Figure 3.* Given $\mathbf{x}$, a MixCEM predicts concepts $\hat{\mathbf{p}}$ and task labels $\hat{\mathbf{y}}$. It achieves this by (1) learning *global* concept embeddings $\bar{\mathbf{c}}^{(+)}, \bar{\mathbf{c}}^{(-)} \in \mathbb{R}^{k \times m}$ and *residual* embeddings $r^{(+)}(\mathbf{x}), r^{(-)}(\mathbf{x}) \in \mathbb{R}^{k \times m}$ for each training concept $c_i$, (2) using these embeddings to estimate $\hat{p}_i = \mathbb{P}(c_i = 1 \mid \mathbf{x}, \bar{C})$ and to construct *contextual* concept embeddings $(\mathbf{c}_i^{(+)}, \mathbf{c}_i^{(-)})$, (3) mixing contextual embeddings to produce a single embedding $\hat{\mathbf{c}}_i$, and (4) predicting $\hat{\mathbf{y}}$ from the bottleneck $\hat{\mathbf{c}} = [\hat{\mathbf{c}}_1, \cdots, \hat{\mathbf{c}}_k]^T$.

dings to avoid being poisoned when $\mathbf{x}$ is OOD. We achieve this in two ways. First, before mixing these embeddings, we adjust the magnitude of the residual component so that it loses its influence when the sample is likely OOD. We do this by scaling the residuals inversely proportionally to their concept prediction's uncertainty, or entropy $H(\hat{p}_i)$: $\mathbf{c}_i^{(+/-)} := \bar{\mathbf{c}}_i^{(+/-)} + \left(1 - H(\hat{p}_i)\right) r_i^{(+/-)}(\mathbf{x})$. As the entropy $H$ of a Bernoulli r.v. is in $[0, 1]$, and it increases if uncertainty is higher, the scaling factor $\left(1 - H(\hat{p}_i)\right)$ controls leakage as a function of concept uncertainty. Second, after the MixCEM is trained, we use Platt scaling (Platt et al., 1999) to *calibrate* its concept predictions $\hat{\mathbf{p}}$ to capture better the model's true uncertainty (see Appendix A for details).

Intuitively, one can think of $\bar{\mathbf{c}}_i^{(+)}$ and $\bar{\mathbf{c}}_i^{(-)}$ as *priors* representing what we know about the implications of concept $c_i$ being "active" or "inactive", respectively. Under this interpretation, the residuals $r_i^{(+)}(\mathbf{x})$ and $r_i^{(-)}(\mathbf{x})$ can be thought of as *evidence* that will enable us to perform posterior updates to these priors after we observe task-relevant information missing from the concept annotations.

**Task Likelihood** Given contextual embeddings $(\mathbf{c}_i^{(+)}, \mathbf{c}_i^{(-)})$, we build a *concept bottleneck* $\hat{\mathbf{c}}$ from where we estimate the task likelihood $\mathbb{P}(y \mid \mathbf{x})$ using a (linear) label predictor model $\hat{\mathbf{y}} = f(\hat{\mathbf{c}})$. We construct this bottleneck by first building a single concept representation $\hat{\mathbf{c}}_i$ for each concept, which we then concatenate into a single bottleneck vector $\hat{\mathbf{c}} := [\hat{\mathbf{c}}_1, \cdots, \hat{\mathbf{c}}_k]^T$. As in CEMs, we do this by mixing contextual embeddings $\mathbf{c}_i^{(+)}, \mathbf{c}_i^{(-)}$ using the predicted concept probability $\hat{p}_i$ as a mixing coefficient: $\hat{\mathbf{c}}_i := \hat{p}_i \mathbf{c}_i^{(+)} + (1 - \hat{p}_i) \mathbf{c}_i^{(-)}$. That way, $\hat{\mathbf{c}}_i$ is closer to $\mathbf{c}_i^{(+)}$ if $c_i$ is predicted to be active and closer to $\mathbf{c}_i^{(-)}$ otherwise.

**Intervening** At test time, we can intervene on $c_i$ by forc-

ing $\hat{p}_i$ to its ground-truth value when computing $\hat{\mathbf{c}}_i$ (e.g., if concept $c_i$ is active, then the expert sets $\hat{p}_i := 1$). This results in $\hat{\mathbf{c}}_i$ becoming $\mathbf{c}_i^{(+)}$ if $c_i$ is active and $\mathbf{c}_i^{(-)}$ otherwise.

**Training Objective** Given a task-specific loss $\mathcal{L}_{\text{task}}(y, \hat{y})$ (e.g., cross-entropy), we train MixCEM by minimising:

$$\mathbb{E}_{(\mathbf{x}, \mathbf{c}, y) \sim \mathcal{D}} \Big[ \mathcal{L}_{\text{task}}\big(y, f(g(\mathbf{x}))\big) + \lambda_c \text{BCE}(\mathbf{c}, \hat{\mathbf{p}}) + \lambda_p \mathcal{L}_{\text{task}}\big(y, f(\bar{\mathbf{c}})\big) \Big]$$

As in jointly trained Vanilla CBMs, the first term here is the task accuracy, while the second term is the mean binary cross-entropy between concept labels and predicted scores. The hyperparameter $\lambda_c \in \mathbb{R}_0^+$ controls how much weight we give to correctly predicting concept labels vs task labels.

The third term, which we call the *prior error* and scale it by $\lambda_p \in \mathbb{R}_0^+$, maximises the task accuracy when *only* the global embeddings are used. Here, $\bar{\mathbf{c}}$ represents the bottleneck formed by mixing the global concept embeddings using the **ground-truth** concept labels as coefficients:

$$\bar{\mathbf{c}} := \Big[ \big(c_1 \bar{\mathbf{c}}_1^{(+)} + (1 - c_1) \bar{\mathbf{c}}_1^{(-)}\big), \cdots, \big(c_k \bar{\mathbf{c}}_k^{(+)} + (1 - c_k) \bar{\mathbf{c}}_k^{(-)}\big) \Big]^T$$

This term maximises the information about the downstream task $y$ encoded in the global concept embeddings. In Appendix B, we prove that *MixCEM's objective function naturally arises* as the MLE of a probabilistic graphical model.

**Non-deterministic Fallbacks** We incentivise MixCEMs to be more receptive to interventions in two ways. First, we randomly set (i.e., intervene on) $\hat{p}_i$ to its ground truth value $c_i$ with probability $p_{\text{int}}$ during training (we use $p_{\text{int}} = 0.25$). We follow this procedure as it was shown to improve intervenability in CEMs (Espinosa Zarlenga et al., 2022), and discuss its importance in Section 6. Second, to enable MixCEMs to handle dropping arbitrary residual concept embeddings, we zero the residual $r_i^{(+/-)}$ with probability $p_{\text{drop}} \in [0, 1]$ during training. As in Dropout (Srivastava

*Table 1.* Task accuracy and mean concept ROC-AUC reported as mean ± stds (%) across three seeds. Each task's best result, and those not significantly different from it (paired $t$-test, $p = 0.05$), are underlined. Note that, as outlined in Section 5.1, we do not expect MixCEM to attain the best result for these metrics. However, we want it to remain competitive against other powerful baselines.

| Method | CUB | CUB-Incomplete | AwA2 | AwA2-Incomplete | CIFAR10 | CelebA |
|---|---|---|---|---|---|---|
| DNN | $71.18_{\pm0.67}$ / N/A | $71.42_{\pm0.30}$ / N/A | $89.20_{\pm0.26}$ / N/A | $89.33_{\pm0.22}$ / N/A | $80.79_{\pm0.22}$ / N/A | $25.39_{\pm0.49}$ / N/A |
| Vanilla CBM | $70.97_{\pm0.76}$ / $89.80_{\pm0.14}$ | $56.46_{\pm0.48}$ / $88.15_{\pm0.14}$ | $87.52_{\pm0.41}$ / $94.42_{\pm0.16}$ | $76.28_{\pm0.82}$ / $93.25_{\pm0.30}$ | $76.97_{\pm0.17}$ / $73.99_{\pm0.20}$ | $24.18_{\pm0.65}$ / $80.12_{\pm0.21}$ |
| Hybrid CBM | $73.65_{\pm0.23}$ / $94.53_{\pm0.04}$ | $72.13_{\pm0.57}$ / $88.97_{\pm0.20}$ | $88.18_{\pm0.65}$ / $94.42_{\pm0.19}$ | $89.39_{\pm0.18}$ / $95.81_{\pm0.01}$ | $79.12_{\pm0.27}$ / $73.78_{\pm0.08}$ | $35.43_{\pm0.23}$ / $87.66_{\pm0.18}$ |
| ProbCBM | $68.16_{\pm1.44}$ / $89.15_{\pm0.65}$ | $60.56_{\pm1.11}$ / $89.26_{\pm0.18}$ | $85.34_{\pm0.39}$ / $94.09_{\pm0.28}$ | $67.12_{\pm0.18}$ / $94.07_{\pm0.28}$ | $64.80_{\pm5.15}$ / $72.08_{\pm0.84}$ | $31.74_{\pm0.29}$ / $88.14_{\pm0.22}$ |
| P-CBM | $69.00_{\pm0.69}$ / $63.22_{\pm0.23}$ | $48.88_{\pm10.66}$ / $61.64_{\pm0.59}$ | $90.31_{\pm0.12}$ / $85.39_{\pm0.23}$ | $75.77_{\pm0.44}$ / $85.78_{\pm0.55}$ | $79.86_{\pm0.05}$ / $72.16_{\pm0.00}$ | $17.18_{\pm2.47}$ / $76.49_{\pm1.12}$ |
| Residual P-CBM | $71.84_{\pm0.57}$ / $63.35_{\pm0.17}$ | $70.69_{\pm0.21}$ / $61.70_{\pm0.64}$ | $90.60_{\pm0.14}$ / $85.39_{\pm0.23}$ | $89.56_{\pm0.17}$ / $85.77_{\pm0.56}$ | $79.90_{\pm0.10}$ / $72.16_{\pm0.00}$ | $15.43_{\pm1.91}$ / $76.49_{\pm1.12}$ |
| CEM | $76.67_{\pm0.11}$ / $89.60_{\pm0.11}$ | $74.42_{\pm0.46}$ / $89.35_{\pm0.19}$ | $91.07_{\pm0.24}$ / $94.84_{\pm0.43}$ | $90.12_{\pm0.07}$ / $96.04_{\pm0.07}$ | $80.05_{\pm0.35}$ / $73.67_{\pm0.40}$ | $34.89_{\pm0.46}$ / $87.79_{\pm0.21}$ |
| IntCEM | $73.33_{\pm0.70}$ / $84.34_{\pm0.53}$ | $72.61_{\pm0.21}$ / $88.02_{\pm0.43}$ | $89.52_{\pm0.92}$ / $84.28_{\pm1.01}$ | $88.65_{\pm0.38}$ / $95.10_{\pm0.09}$ | $78.48_{\pm0.68}$ / $66.79_{\pm0.30}$ | $36.93_{\pm1.07}$ / $88.08_{\pm0.16}$ |
| MixCEM (ours) | $76.54_{\pm0.14}$ / $88.00_{\pm0.44}$ | $74.54_{\pm0.19}$ / $87.24_{\pm0.43}$ | $89.94_{\pm0.12}$ / $93.35_{\pm0.04}$ | $88.68_{\pm0.05}$ / $95.19_{\pm0.02}$ | $78.64_{\pm0.41}$ / $72.52_{\pm0.69}$ | $35.58_{\pm0.72}$ / $87.51_{\pm0.12}$ |

et al., 2014), this can be seen as learning an ensemble of models where each model includes only a subset of the residuals. At inference, we adjust for this effect by sampling $M$ bottlenecks (we fix $M = 50$ in practice) and averaging the prediction made from all $M$ bottlenecks similarly to how Monte Carlo Dropout operates (Gal & Ghahramani, 2016). This mechanism has the added benefit of mitigating overfitting and further preventing poisonous leakage. For a thorough ablation of MixCEM's hyperparameters showing its robustness across values, see Appendix J.

## 5. Experiments

**Research Questions** We explore the following questions:

**(Q1)** Do MixCEMs have a high concept and task fidelity?
**(Q2)** Do MixCEMs remain intervenable and correctly bounded for ID and OOD samples?
**(Q3)** Are MixCEMs robust to OOD shifts?
**(Q4)** Do MixCEM's bottlenecks go OOD for OOD inputs?

**Datasets** We study these questions on the following tasks: (1) CUB (Wah et al., 2011), a bird classification task with 200 classes and 112 concepts selected by Koh et al. (2020), (2) AwA2 (Xian et al., 2018), an animal classification task with 50 classes and 85 concepts, (3) CelebA (Liu et al., 2018), a face recognition task with 256 classes and 6 concepts selected by Espinosa Zarlenga et al. (2022), and (4) CIFAR-10 (Krizhevsky et al., 2009), a classification task with 10 classes and with 143 concepts obtained in an unsupervised manner by Oikarinen et al. (2023). Finally, we construct concept-incomplete versions of CUB and AwA2 by randomly selecting 25% and 10% of their concepts, respectively. All datasets are described in Appendix C.

**Baselines** We compare MixCEMs against Vanilla CBMs (Koh et al., 2020), Hybrid CBMs (Mahinpei et al., 2021), CEMs (Espinosa Zarlenga et al., 2022), IntCEMs (Espinosa Zarlenga et al., 2023a), ProbCBMs (Kim et al., 2023), and P-CBMs (including their residual version) (Yuksekgonul et al., 2023). Moreover, we include a vanilla DNN as a representative black-box baseline. When possible, all baselines are given the same capacity and budget for fine-tuning and training. We select hyperparameters based on the area under the validation task-accuracy vs intervention curve and describe all hyperparameters and architectures in Appendix D. Finally, for Vanilla CBMs, here we focus on their sigmoidal and jointly trained versions. However, we discuss results for different variants (e.g., sequential, independent, and logit CBMs) in Appendix E.

We note that we do not explicitly include Label-free CBMs (Oikarinen et al., 2023) in our evaluation, although we do use their labelling procedure to obtain concept labels in CIFAR10, as in datasets where we have concept annotations (e.g., CUB), it is difficult to fairly compare Label-free CBMs and concept-supervised methods without ground-truth labels for label-free concepts. Moreover, we do not include GlanceNets (Marconato et al., 2022) in our evaluation, even though these models use a leakage/OOD detector, because once leakage is detected by GlanceNet's OOD detector, the model does not provide a solution that allows operations like interventions to work in that instance.

### 5.1. Task and Concept Fidelity (Q1)

We first study MixCEM's task and concept fidelity through its task accuracy and mean concept ROC-AUC. As we are interested in designing models that satisfy both CA and BI, we emphasise that we *do not* expect MixCEM to be the best-performing baseline in terms of its ID task and concept fidelity. Nonetheless, we use this study to verify that MixCEM's task and concept performances are competitive against existing methods. Our results, summarised in Table 1 and discussed below, suggest that this is indeed the case.

**MixCEM is completeness-agnostic and attains competitive task accuracies (Table 1, red).** In three out of six tasks, MixCEM is in the set of best-performing baselines with respect to task accuracy. We note that, in tasks where it underperforms, MixCEM's drop against the best-performing CM (i.e., CEM) is relatively small (at worst close to 2% difference). More importantly, noticing that MixCEM's accuracy outperforms or closely matches that of black-box DNNs on concept-complete *and* incomplete

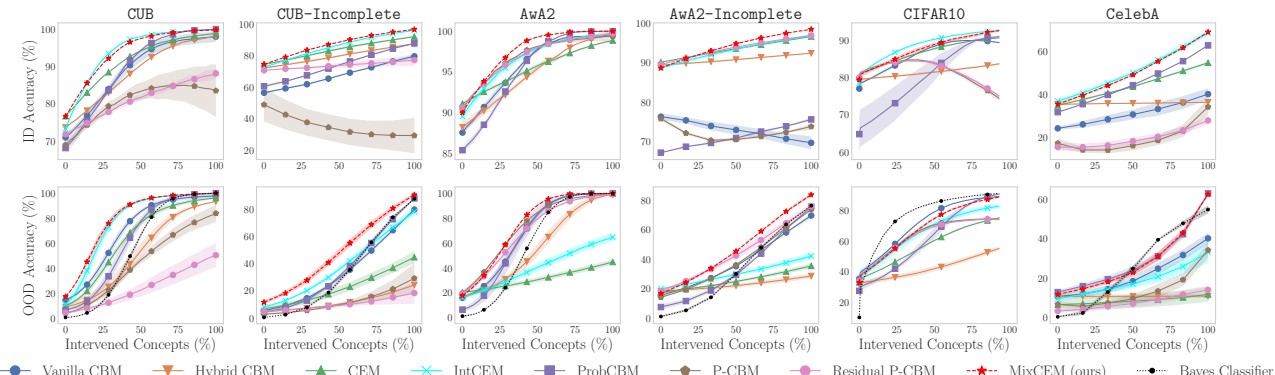

*Figure 4.* Task accuracy as we intervene on concepts, selected at random, for ID (top) and OOD (bottom) test samples. OOD samples have a form of salt & pepper noise injected into at most 10% of their channels (similar results on other forms of distribution shifts can be seen in Appendix H). For the sake of efficiency, in the OOD plots we approximate a Bayes classifier that takes as an input only the intervened concepts using a masked MLP (see Appendix D.4 for further details). Notice that across all tasks, both in ID and OOD instances, the area under the intervention curve for MixCEM is higher than that of competing approaches. See Appendix M for a tabulation of these results, including the estimated area under each intervention curve.

tasks, our results indicate that MixCEMs are completeness-agnostic. The same does not hold for Vanilla CBMs, P-CBMs, and ProbCBMs.

**MixCEM has a slight drop in concept AUC, yet it remains competitive with respect to state-of-the-art CMs (Table 1, blue).** MixCEM's mean concept ROC-AUC is slightly below the best-performing model in four of our tasks. Nevertheless, MixCEM's drop in mean concept ROC-AUC is relatively small for most tasks ($< 2\%$) and consistently within the performance of similar SotA baselines, such as CEMs or IntCEMs. The only task exception is in CUB, where Hybrid CBMs attain significantly higher scores. However, as discussed next, Hybrid CBMs are not generally intervenable. Therefore, they are not suitable candidates for human-in-the-loop scenarios, our setup of interest.

### 5.2. Intervenability (Q2)

We evaluate MixCEM's intervenability and show it is bounded for OOD samples. For this, we look at a model's task accuracy as we perform concept interventions. As in previous works, we select intervened concepts uniformly at random and intervene on groups of related concepts simultaneously when these groups are known (e.g., in CUB).

**When intervened on for ID samples, MixCEM outperforms competing baselines (Figure 4, top).** Our results show that MixCEMs not only significantly improve their task accuracy the more one intervenes (i.e., they are *intervenable*), but they perform better or on par with IntCEMs (the best-performing baseline here). More crucially, they achieve this without expensive training-time sampling, leading to faster training times than IntCEMs (see Appendix F). Finally, we observe that MixCEM's interventions significantly improve its performance in concept-incomplete tasks, suggesting that its embeddings properly leak information even after interventions. This is in contrast with non-leaky approaches (e.g., Vanilla CBMs and P-CBMs) and global-embedding-based approaches (e.g., ProbCBMs), which significantly underperform in concept-incomplete tasks.

**MixCEM achieves bounded intervenability and better OOD intervention accuracy (Figure 4, bottom).** Figure 4 shows the results of intervening on test samples corrupted with a form of "Salt & Pepper" noise. We study this form of noise as it is common within real-world deployment (Hendrycks & Dietterich, 2019; Mousavi et al., 2017) (see examples of corrupted images in Appendix G). However, we emphasise that, as shown in Appendix H, MixCEM's intervention improvements discussed below are also seen for other forms of real-world distribution shifts. This includes distribution shifts caused by downsampling, blurring, random affine transformations, and domain shifts (e.g., a model trained with MNIST (Deng, 2012) digits is intervened on samples containing real-world colour digits).

When looking at interventions on OOD samples, we see that MixCEM is the *only* completeness-agnostic baseline whose interventions are usually bounded: all of MixCEM's OOD intervention curves, except in CIFAR10 and a short instance in CelebA, are always near or above the accuracy of the Bayes Classifier (BC, black dashed line). We believe MixCEM's underperformance with respect to the BC in CIFAR10 and CelebA results from concepts in both datasets being difficult to properly learn for all methods due to concept label noise (their concept annotations come from CLIP-based classification or subjective human annotations, both prone to mistakes). Nevertheless, across *all* tasks, MixCEMs have significantly higher OOD intervention ac-

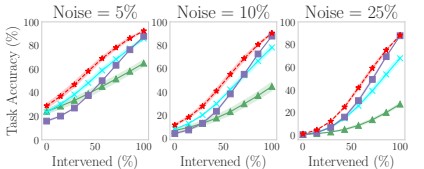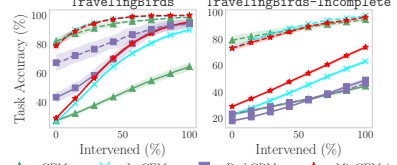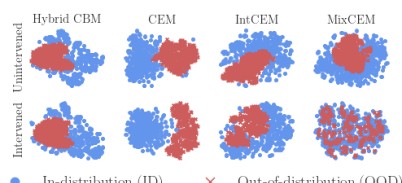

*Figure 5.* **(Left)** `CUB-Incomplete` intervention curves. Test samples are perturbed by adding "Salt & Pepper" noise with increasing levels (% pixels corrupted). **(Centre)** `TravelingBirds` intervention curves. We show results on a spuriously correlated validation set (dashed) and a test set without the spurious correlation (solid). **(Right)** t-SNE projections of bottlenecks before (top) and after (bottom) all concepts are intervened on for ID and OOD samples in `CUB-Incomplete`.

curacies than competing completeness-agnostic baselines, especially CEMs and IntCEMs (up to approx. 48% and 41% improvement in `AwA2-Incomplete` over CEM and IntCEM, respectively). Finally, the fact that MixCEM even surpasses the BC when all concepts are intervened on in concept-incomplete tasks (e.g., `AwA2-Incomplete` and `CelebA`) suggests that MixCEMs can exploit useful leakage for OOD samples while avoiding leakage poisoning.

### 5.3. Unintervened OOD Robustness (Q3)

Next, we study MixCEM's performance across different distribution shifts. In particular, we evaluate MixCEMs as we (1) vary the amount of test noise on `CUB-Incomplete`, and (2) train it on the `TravelingBirds` dataset (Koh et al., 2020), a variation of `CUB` where a training-time spurious correlation is introduced between the background and the downstream task labels (see Appendix C for examples). Below, we discuss how our results in Figure 5 suggest that MixCEM exhibits better OOD robustness. For simplicity, we focus on the best-performing baselines. However, we show our observations extend to all baselines in Appendix I.

**With and without interventions, MixCEM is more robust to test-time noise (Figure 5, left).** Our noise ablation for `CUB-Incomplete` (Figure 5, left) shows that across noise levels, MixCEM achieves better task accuracy as it is intervened on than our baselines. More importantly, we see that for low-to-medium noise regimes (e.g., 5% and 10%), MixCEM achieves the best unintervened performance (leftmost points). For example, when noise = 5%, MixCEM's unintervened task accuracy is 29.69% vs CEM's 24.08%. This trend is observed for incomplete *and* complete tasks (see `CUB` results in Appendix I.1). These results suggest that MixCEMs are robust models even without interventions.

**MixCEM is more robust to spurious correlations in concept-incomplete tasks (Figure 5, centre).** In Figure 5 (centre), we look at how training-time spurious correlations affect MixCEM's performance in concept-complete and incomplete versions of `TravelingBirds`. In the incomplete task, we see that MixCEM's unintervened and intervened accuracies are better than our baselines', both

for test sets with the spurious correlation (ID samples) and without it (OOD samples). In particular, MixCEM's intervened performance is drastically better than that of IntCEMs when the spurious correlations disappear (adding more than 10% points in task accuracy when all concepts are intervened on). Nevertheless, we also notice that in concept-complete setups, although MixCEMs outperform IntCEMs once more, they are outperformed by ProbCBMs when no interventions are made on the test set without the spurious correlation (OOD samples). In Section 6, we provide some intuition as to why this may be the case. Regardless, when compared against other completeness-agnostic approaches (e.g., Residual P-CBMs, CEMs, IntCEMs, Hybrid CBMs), our results suggest that MixCEMs are more robust to different forms of OOD shifts.

### 5.4. Concept Bottleneck Analysis (Q4)

Finally, we qualitatively study how concept bottlenecks are affected by concept interventions when samples go OOD. For this, we look at the t-SNE (Van der Maaten & Hinton, 2008) projections of concept bottlenecks for different CMs. Our experiments' conclusions are described below.

**MixCEM's bottlenecks remain within distribution for both ID and OOD samples (Figure 5, right).** In Figure 5 (right), we see that in contrast to bottlenecks in CEMs and Hybrid-CBMs, MixCEM's unintervened bottlenecks (top of figure) appear to remain within their ID bottleneck distribution for OOD samples. More importantly, they appear to closely match their original distribution when all concepts are intervened. From our baselines, only IntCEM's bottlenecks seem to remain closer to their original distribution after all interventions. This may explain why IntCEM's OOD interventions outperform CEM's. However, we observe that MixCEM's bottlenecks capture a significantly greater proportion of the variance in their ID bottleneck distribution than IntCEM's, potentially explaining why OOD interventions are more effective in MixCEM. These results suggest MixCEM learns concept representations that remain within distribution and avoid leakage poisoning for both ID and OOD samples.

# 6. Discussion and Conclusion

**Prior Optimisation and Intervention Awareness**  Our experiments hint at a relationship between OOD intervention robustness and intervention awareness: IntCEM, which has an intervention-aware loss, and MixCEM, which has a robustness-aware loss minimising a prior's error $\mathcal{L}_{\text{task}}\big(y, f(\bar{\mathbf{c}})\big)$, achieve the best intervention accuracies for ID samples across all baselines. This suggests that robustness awareness can have a positive effect on ID intervenability. In the case of MixCEM, we believe its prior error minimisation leads to both better ID and OOD intervenability because this term has an implicit incentive to maximise the model's performance both when all concepts are intervened on (as $\bar{\mathbf{c}}$ is constructed by mixing embeddings based on the ground-truth concept labels) and when no leakage is allowed. Hence, this term has the extra effect of penalising MixCEM for mispredicting $y$ when *all* concepts are intervened on. This has three surprising results: (1) MixCEM's ID intervenability matches IntCEM's, *without* needing a complex IntCEM-like sampling-based loss; (2) even when we set the concept weight loss to zero (i.e., $\lambda_c = 0$), MixCEMs remain highly intervenable as long as $\lambda_p > 0$ (see Appendix J.1); and (3) improvements from training-time interventions (i.e., when $p_{\text{int}} > 0$) are much smaller for MixCEM than what has been observed for CEMs (see Appendix J.5). All of these suggest that prior error minimisation serves as a robust *intervention-aware regulariser*.

**Learning Intervention Policies In MixCEM**  As Mix-CEMs are embedding-based methods, we can utilise the intervention-aware training pipeline used in IntCEMs to learn an intervention policy as a by-product of training. In Appendix K we evaluate this variant of MixCEM, which we call "*MixIntCEM*", for the CUB-based datasets. There, we observe four interesting results. First, MixIntCEMs remain as accurate, without any interventions, as comparable IntCEMs and MixCEMs. Second, MixIntCEM learns a highly effective intervention policy that significantly outperforms a random intervention policy on the original Mix-CEM, both for complete and incomplete tasks. Next, we see that MixIntCEM's OOD intervenability remains properly bounded and high (almost identical to the original Mix-CEM when interventions are selected randomly). More importantly, however, we see that MixIntCEM's intervention policy yields better intervention curves than all other baselines for OOD test sets. Finally, we nonetheless observe that, for some instances, intervening on IntCEM using its learnt policy yields better intervention curves than those seen when we intervene on MixIntCEM using its own learnt policy. Nevertheless, we observe this only when intervening on ID test sets for complete datasets, suggesting there is still some value in maintaining the original IntCEM pipeline if we know we are operating in a complete task where test

samples will not go OOD. Considered together, all of these results strongly suggest that IntCEM's intervention learning and MixCEM's embedding decomposition serve as **general design principles** that can be applied to build better, more intervenable CMs depending on the circumstances under which one expects the CM to be deployed.

**Bias Mitigation**  Our `TravelingBirds` results suggest that models incorporating constant global embeddings (e.g., MixCEMs and ProbCBMs) better deal with spurious correlations. We believe this is because global embeddings, by definition, block the flow of concept-independent information during inference. Thus, the model must learn to operate with the same shared representations for samples with and without the spurious correlation. This leads to models learning representations that better capture under-represented groups (e.g., samples without a spurious correlation) and may explain why ProbCBM, built on top of global embeddings, achieves a high OOD uninterved accuracy in `TravelingBirds`. Future work could then explore how global embeddings can be exploited for generalisation.

**Limitations**  MixCEMs require more parameters (i.e., $\mathcal{O}(km)$ weights for $\bar{C}$) and hyperparameters (e.g., $\lambda_p$ and $p_{\text{drop}}$) than CEMs. Although MixCEM is generally robust to its hyperparameters (see Appendix J), its memory and fine-tuning footprint open the door for future work to alleviate these constraints. Moreover, we foresee at least two potential failure modes in MixCEMs: First, when a concept goes OOD and the shift renders the concept incomprehensible for an expert, MixCEMs may fail to completely block leakage poisoning as one cannot intervene on such a concept. Hence, future work can explore mechanisms for blocking all unwanted leakage without knowing a concept's label. Second, in incomplete tasks, intervened MixCEMs do not always recover the full ID performance in OOD inputs. Therefore, future work can explore how information about unprovided concepts can be better preserved after an intervention. Finally, future work could explore (1) extending MixCEM's embedding decomposition to other embedding-based methods, such as ECBMs, and (2) devising better ways to inject priors into its global embeddings.

**Conclusion**  In this paper, we show that previous state-of-the-art concept-based models are ill-equipped to concurrently handle both concept-incompleteness and test-time interventions when inputs are OOD. To address this, we introduce MixCEM, a new concept-based architecture that uses an entropy-based gating mechanism to control when and how concept-independent feature information is leaked. Through an extensive evaluation across concept-complete and concept-incomplete tasks, we show that MixCEMs outperform strong baselines by significantly improving accuracy for both ID and OOD samples in the presence and absence of concept interventions.

## Impact Statement

This paper presents work whose goal is to advance the field of Machine Learning. There are many potential societal consequences of our work, none of which we feel must be specifically highlighted here.

## Acknowledgements

MEZ acknowledges support from the Gates Cambridge Trust via a Gates Cambridge Scholarship. GD acknowledges support from the European Union's Horizon Europe project SmartCHANGE (No. 101080965), TRUST-ME (No. 205121L_214991) and from the Swiss National Science Foundation projects XAI-PAC (No. PZ00P2_216405). PB acknowledges support from the Swiss National Science Foundation project IMAGINE (No. 224226). MJ acknowledges the support of the U.S. Army Medical Research and Development Command of the Department of Defense; through the FY22 Breast Cancer Research Program of the Congressionally Directed Medical Research Programs, Clinical Research Extension Award GRANT13769713. Opinions, interpretations, conclusions, and recommendations are those of the authors and are not necessarily endorsed by the Department of Defense.

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

# A. MixCEM Concept Probability Calibration

Platt scaling (Platt et al., 1999) is a post-hoc calibration method used to transform the outputs of a probabilistic classifier into well-calibrated probabilities (i.e., probabilities that better represent the model's true uncertainty). In this work, we apply a common adaptation of Platt Scaling by Guo et al. (2017) to calibrate a MixCEM's concept probabilities so that they better represent their uncertainty and, therefore, may serve as better indicators for when a concept might have gone OOD.

In practice, Platt scaling involves fitting a logistic regression model to the logits (pre-sigmoid activations) of MixCEM's concept scores $\hat{\mathbf{p}}$ on a validation dataset. Specifically, for each concept $c_i$ in our training set, given the logit $z_i := \sigma^{-1}(\hat{p}_i)$ of concept prediction $\hat{p}_i$, we calibrate its output by learning a linear transformation of $z_i$:

$$\mathbb{P}(c_i = 1 \mid z_i) = \sigma\big(a_i z_i + b_i\big),$$

where $a_i, b_i \in \mathbb{R}$ are parameters specific to each concept. These parameters are learnt using maximum likelihood estimation for $E_{\text{cal}}$ epochs on the **validation data** after the MixCEM has been trained.

When we learn parameters $\mathbf{a}$ and $\mathbf{b}$, we freeze all other parameters in MixCEM and minimise the Binary Cross Entropy loss between scaled concept predictions and their ground-truth labels in the validation set. This means that the model's concept validation accuracy remains the same throughout this process, although the task accuracy may change slightly (as the task predictions are a function of the concept predictions). By the end of this optimisation, MixCEM's concept predictions are a better fit to represent the true model uncertainty and can therefore be better at identifying when a sample's concepts have gone OOD. The results of including Platt Scaling as a post-processing step of MixCEMs are further discussed in Appendix J.4.

# B. Maximum likelihood of MixCEM

Once we have constructed MixCEM's global, residual, and contextual embeddings, we need to devise a proper training objective that captures all our intended desiderata. In this section, we derive such an objective by framing MixCEM as a graphical model and deriving a likelihood function, the maximisation of which will enable us to learn our model's parameters given the concept and task likelihoods we derived above.

To construct our training objective, we note that most CMs implicitly assume that there exists a *complete* set of concepts $C^*$ that are the generating factors of variation of the samples $X$ and the downstream tasks $Y$ (see Figure 6a). When we train a CM such as CBM, however, we operate under the assumption that we observe features $X$ and, from these features, we need to infer a potentially *incomplete* subset of concepts $C$ (i.e., the training concepts) from which we then infer a downstream label $Y$ (see Figure 6b). Notice therefore that because $C$ may not contain all the information found initially in $C^*$, traditional CBMs may struggle to achieve CA unless additional information bypasses or channels are introduced in their inference process. Therefore, we will frame MixCEM as a graphical model that explicitly represents the fact that the set of training concepts $C$ will require some additional residual information $R$ for us to properly reconstruct a complete set of concepts $C^*$ from which we can accurately predict $Y$.

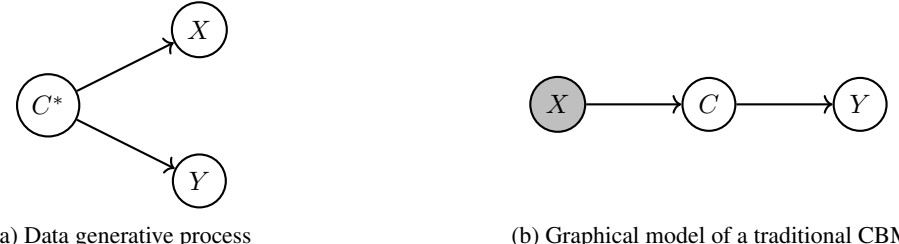

(a) Data generative process          (b) Graphical model of a traditional CBM

*Figure 6.* Assumptions underlying CMs. (a) Generative process assumed by CMs, and (b) graphical model underlying CBMs, where darkened nodes represent observed variables (i.e., inputs). The implied generative process for CMs assumes the existence of a set of *complete* concepts $C^*$ that can perfectly describe features $X$ and task labels $Y$. In contrast, traditional CMs such as CBMs operate under the assumption that, given a set of observable input features $X$, they must first infer a *potentially incomplete* concept set $C$ (i.e., the training concept set) and then they must infer a downstream task label $Y$ from those concepts.

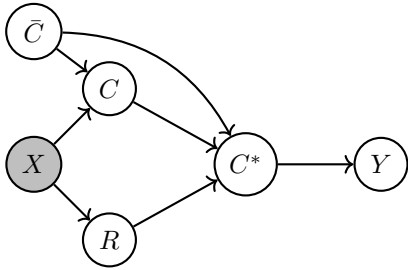

*Figure 7.* MixCEM's graphical model. MixCEM assumes that a complete set of concepts $C^*$ can be constructed from (1) a set of concepts aligned with the training concept annotations $C$, that follow some prior $\bar{C}$, and (2) a set of residual variables $R$ that describe all information missing from $C$ that is relevant for predicting the downstream task $Y$.

In our framing of MixCEM in Section 4, we argued that we can decompose our concept activations $C$ into two components: one that is sample-specific (i.e., the residual embeddings) and one that is concept-specific (i.e., the global embeddings). Without loss of generality, we can formally express this consideration in a single graphical model (see Figure 7) by letting the complete concept set $C^*$ be a function of three factors: two independent factors $R$ and $\bar{C}$ and a third dependent factor $C$. Here, (1) $C$ represents a potentially incomplete set of training concepts given to MixCEM, (2) $\bar{C}$ represents a prior set of beliefs about concepts in $C$ (it is independent of a specific input observation), and (3) $R$ represents all residual information in $X$ that is captured by the complete concept set $C^*$ but is missing from the training concepts $C$. Hence, $R$ provides the appropriate context and a higher level of detail about the input observation when predicting $Y$ from a set of incomplete concepts $C$.

Assuming we observe $X$, MixCEM's graphical model can be factorised as follows:

$$\mathbb{P}(Y, C \mid \bar{C}, X) = \sum_R \sum_{C^*} \mathbb{P}(Y \mid C^*)\mathbb{P}(C^* \mid C, \bar{C}, R)\mathbb{P}(C \mid X, \bar{C})\mathbb{P}(R \mid X) \tag{1}$$

where:

- $\mathbb{P}(Y \mid C^*)$ is a categorical distribution over task labels given the complete concepts $C^*$. In MixCEM, we parameterise this model via the label predictor classifier $f : C^* \to Y$.

- $\mathbb{P}(C \mid X, \bar{C})$ is a Bernoulli distribution that models the presence or absence of each concept $c_i$ in our concept set $\mathbf{c} \sim C$ given $X$ and $\bar{C}$. In MixCEM, this distribution is parameterised by the concept scoring functions $\{s_i\}_{i=1}^k$, where we interpret $\bar{C}$ as learnable parameters of the classifier corresponding to MixCEM's global embeddings.

- $\mathbb{P}(C^* \mid C, \bar{C}, R)$ is a *deterministic* bottleneck encoder $\gamma : C \times R \times \bar{C} \to C^*$ generating MixCEM's bottleneck $\hat{\mathbf{c}}$ from the concept values, the priors, and the residuals. As both the concepts and the residuals are functions of $\mathbf{x}$, we express this bottleneck as $\gamma(\mathbf{x}, \bar{\mathbf{C}})$.

- $\mathbb{P}(R \mid X)$ is a *deterministic* function capturing MixCEM's residuals $\{(r_i^{(+)}(\mathbf{x}), r_i^{(-)}(\mathbf{x})\}_{i=1}^k$. Abusing notation, we encapsulate this learnable function as $r : X \to R$.

We can use this factorisation to learn MixCEM's parameters. Specifically, given a concept-annotated dataset of i.i.d. triples $\mathcal{D} = \{(\mathbf{x}^{(j)}, \mathbf{c}^{(j)}, y^{(j)})\}_{j=1}^N$, MixCEM's parameters can be learnt via gradient descent by maximising the empirical log-likelihood of the training data. For this, we consider the prior variables $\bar{\mathbf{C}} \sim \bar{C}$ as parameters for the optimisation, similar to the weights $\theta_g$ and $\theta_f$, which represent the weights of the concept encoder and label predictor, respectively. This yields the following objective function:

$$\theta_g^*, \theta_g^*, \bar{\mathbf{C}}^* = \underset{\theta_g, \theta_f, \bar{\mathbf{C}}}{\arg\max} \, \mathbb{E}_{(\mathbf{x}, \mathbf{c}, y) \sim \mathcal{D}}\Big[\mathbb{P}(Y = y, C = \mathbf{c} \mid X = \mathbf{x}; \, \theta_g, \theta_f, \bar{\mathbf{C}})\Big] \tag{2}$$

Considering that both the residual and bottleneck encoder distributions are deterministic functions of their inputs, for any sample $(\mathbf{x}, \mathbf{c}, y) \sim \mathcal{D}$ we can use Equation 1 to rewrite the objective function in the expectation above as:

$$\mathbb{P}(Y = y, C = \mathbf{c} \mid X = \mathbf{x}; \theta_g, \theta_f, \bar{\mathbf{C}}) = \mathbb{P}(Y = y \mid C^* = \gamma(\mathbf{x}, \bar{\mathbf{C}}); \theta_g, \theta_f)\mathbb{P}(C = \mathbf{c} \mid X = \mathbf{x}; \theta_g, \bar{\mathbf{C}})$$

Here, we make a practical modelling assumption. When learning our parameters $\theta_f$, $\theta_g$, and $\bar{\mathbf{C}}$, we want to disentangle the gradients from the downstream task $Y$ that update parameters $\theta_f$ and $\theta_g$ from those that update $\bar{\mathbf{C}}$. This is because we want our prior embeddings $\bar{\mathbf{C}}$ to be, on their own, as informative as possible about the downstream task so that the residuals $r(X)$ only contribute to $C^*$ with information about $Y$ that is not already encoded in $C$. As such, we factorise the likelihood of the downstream task $Y$ with respect to $C^*$ as:

$$\mathbb{P}(Y \mid C^*) = \mathbb{P}\big(Y \mid C, r(X), \bar{C}\big)\mathbb{P}\big(Y \mid \bar{\mathbf{c}}\big) \tag{3}$$

where

- $\mathbb{P}\big(Y \mid C, r(X), \bar{C}\big)$ represents the task distribution when observing the residual embeddings, the predicted training concepts, and the prior embeddings. In other words, this is the task likelihood we wrote down for MixCEM in Section 4.

- $\mathbb{P}\big(Y \mid \bar{\mathbf{c}}\big)$ represents the task distribution given the training concepts alone. This likelihood can be interpreted as the label $f(\bar{\mathbf{c}})$ predicted when we only provide the label predictor $f$ with the global prior embeddings of the training concepts. Therefore, in practice, $\bar{\mathbf{c}}$ can be thought of as the bottleneck formed by mixing the global (prior) concept embeddings using the **ground-truth** concept labels as coefficients:

$$\bar{\mathbf{c}} = \left[ \big(\mathbf{c}_1\bar{\mathbf{c}}_1^{(+)} + (1 - \mathbf{c}_1)\bar{\mathbf{c}}_1^{(-)}\big), \cdots, \big(\mathbf{c}_k\bar{\mathbf{c}}_k^{(+)} + (1 - \mathbf{c}_k)\bar{\mathbf{c}}_k^{(-)}\big) \right]^T \tag{4}$$

This simplification allows us to express MixCEM's log-likelihood $\mathrm{LL}(\mathbf{x}, \mathbf{c}, y) = \log \mathbb{P}(Y = y, C = \mathbf{c} \mid X = \mathbf{x}; \ \theta_g, \theta_f, \bar{C})$ of a data sample $(\mathbf{x}, \mathbf{c}, y)$ as:

$$\mathrm{LL}(\mathbf{x}, \mathbf{c}, y) = \log \left( \mathbb{P}(y \mid C^* = g(\mathbf{x}); \theta_f, \theta_g, \bar{\mathbf{C}})\mathbb{P}(y \mid C^* = \bar{\mathbf{c}}; \theta_f, \theta_g)\mathbb{P}(\mathbf{c} \mid X = \mathbf{x}; \theta_g, \bar{C}) \right)$$

$$= \log \mathbb{P}\big(y \mid C^* = g(\mathbf{x})\big) + \log \mathbb{P}\big(y \mid C^* = \bar{\mathbf{c}}\big) + \log \mathbb{P}\big(\mathbf{c} \mid X = \mathbf{x}\big)$$

Plugging in MixCEM's likelihoods for the task and concept predictions derived in Section 4 and minimising the *negative* log-likelihood, we get a **final training objective**:

$$\underset{\theta_g, \theta_f, \bar{C}}{\arg\min} \ \mathbb{E}_{(\mathbf{x}, \mathbf{c}, y) \sim \mathcal{D}} \left[ - \log \mathbb{P}\big(y \mid g(\mathbf{x})\big) - \log \mathbb{P}\big(y \mid \bar{\mathbf{c}}\big) - \log \mathbb{P}\big(\mathbf{c} \mid \mathbf{x}\big) \right] =$$

$$\underset{\theta_g, \theta_f, \bar{C}}{\arg\min} \ \mathbb{E}_{(\mathbf{x}, \mathbf{c}, y) \sim \mathcal{D}} \left[ \mathcal{L}_t\big(y, f(g(\mathbf{x}))\big) + \lambda_p \mathcal{L}_t\big(y, f(\bar{\mathbf{c}})\big) + \lambda_c \mathrm{BCE}\big(\mathbf{c}, \hat{\mathbf{p}}\big) \right]$$

where $\mathcal{L}_t(y, \hat{\mathbf{y}})$ is a task-specific loss that minimises the negative log likelihood of the downstream task labels $y$ (e.g., cross-entropy) and $\mathrm{BCE}(\mathbf{c}, \hat{\mathbf{p}})$ is the average binary cross-entropy loss between the ground-truth concepts $\mathbf{c}$ and MixCEM's predicted concept probabilities $\hat{\mathbf{p}}$.

Notice that in the final objective loss function, we include two regularisation hyperparameters, $\lambda_c$ and $\lambda_p$, that allow us to control how much each term affects the parameters we learn (we discuss the effect each hyperparameter has in Section 6). As in traditional jointly trained CBMs and CEMs, $\lambda_c \in \mathbb{R}_0^+$ controls how much we weigh concept fidelity vs task fidelity; in other words: how much we weigh the third term of the objective function above. In contrast, our second hyperparameter, $\lambda_p \in \mathbb{R}_0^+$, maximises the task accuracy when *only* the global embeddings are used (i.e., the second term in the objective function). The loss term corresponding to this hyperparameter, which we call the *prior error*, maximises the information about the downstream task $y$ encoded in the global concept embeddings, ensuring that the learnt global embeddings are informative enough to predict the downstream task even when no additional information is leaked from $\mathbf{x}$ into our label predictor. From an information-theoretic perspective, this leads to MixCEM creating an information bottleneck where most of the information about $Y$ is encoded in $\bar{C}$ and only minimal residual information flows from $X$ in the form of the residuals.

**Information theoretic discussion**  In IID cases, if the concept prior $\bar{C}$ is not informative enough to make predictions, the model could exploit the contextual information in $X$ to refine concept beliefs and generate more accurate concept predictions. In this setting, $X$ is useful to attain high task fidelity (ensuring completeness-agnosticism). However, in non-i.i.d. settings, $X$ encodes features from an unknown distribution, which may result in concept posteriors that are worse than the prior. For

this reason, we may need to exclude $X$ from the computation of $Y$ in such cases. If we exclude $X$ from the computation, we need $\bar{C}$ to incorporate as much information as possible about $Y$; otherwise, our predictions will not be better than random chance, and interventions will be ineffective.

From an information theoretic perspective, MixCEM creates an interpretable information bottleneck where most the information about $Y$ is encoded in $\bar{C}$ and only minimal residual information flows from $X$ (i.e., $I(Y; \bar{C}) \gg I(Y; X)$) as the loss term $\mathcal{L}_{\text{task}}\big(y, f(\bar{\mathbf{c}})\big)$ encourages the model to make predictions from global concepts alone.

## C. Datasets

Below, we discuss the tasks and datasets used for our experiments in Section 5. A summary of the main characteristics of each task can be found in Table 2.

*Table 2.* High-level properties of all tasks used in our experiments.

| Dataset | Training Samples ($N$) | Validation Samples | Testing Samples | Input Shape ($n$) | # Labels ($L$) | # Concepts ($k$) | # Concept Groups |
|---|---|---|---|---|---|---|---|
| CUB | 4,796 | 1,198 | 5,794 | (3, 299, 299) | 200 | 112 | 28 |
| CUB-Incomplete | 4,796 | 1,198 | 5,794 | (3, 299, 299) | 200 | 22 | 7 |
| AwA2 | 22,393 | 7,464 | 7,465 | (3, 224, 224) | 50 | 85 | 28 |
| AwA2-Incomplete | 22,393 | 7,464 | 7,465 | (3, 224, 224) | 50 | 9 | 6 |
| CelebA | 11,818 | 1,689 | 3,376 | (3, 64, 64) | 256 | 6 | N/A |
| CIFAR10 | 40,000 | 10,000 | 10,000 | (3, 32, 32) | 10 | 143 | N/A |
| TravelingBirds | 4,796 | 1,198 | 5,794 | (3, 299, 299) | 200 | 112 | 28 |
| TravelingBirds-Incomplete | 4,796 | 1,198 | 5,794 | (3, 299, 299) | 200 | 22 | 7 |

**CUB**  The CUB bird classification image task is constructed from the Caltech-UCSD Birds-200-2011 dataset (Wah et al., 2011). Each sample in this task corresponds to a $(3 \times 299 \times 299)$ RGB image of a bird (normalised in $[0, 1]$), annotated with one of 200 bird species. Here, each image has 312 binary attribute annotations (e.g., "black nape", "yellow wing colour", etc.). We construct a set of 112 binary concepts following the selection of attributes used by Koh et al. (2020). Moreover, we follow the same majority-voting standardisation of concepts across classes as in (Koh et al., 2020). This leads to all samples from the same class having the same concept profiles. We do this so that this task's concept annotations are truly *complete* with respect to the downstream task (i.e., they can fully describe each downstream label) and to maintain this dataset aligned with how it is usually used in the concept-based XAI literature. Finally, all 112 concepts can be grouped into 28 groups that encapsulate semantically related concepts (e.g., "black wing colour" and "red wing colour"). When performing interventions, all concepts within the same group are intervened on at once, meaning we do interventions on a group-level basis. As it is traditionally done for this dataset Koh et al. (2020); Espinosa Zarlenga et al. (2022); Kim et al. (2023), we normalise all images, and during training, we randomly flip and crop images. For this task, and its incomplete version, we use the same train-validation-test splits as in (Koh et al., 2020).

**CUB-Incomplete**  The CUB-Incomplete task is generated from our CUB task by randomly selecting 25% of CUB's concept groups before training and using the labels of concepts within those groups as each sample's concept labels. We do the concept subsampling only once and use the same subselection across all seeds/rounds of experiments. This resulted in us randomly selecting the following 7 groups of concepts {"bill shape", "head pattern", "breast colour", "bill length", "wing shape", "tail pattern", "bill colour"}. Together, all of these concept groups yield a total of 22 binary concept annotations. We use this dataset as an example of a concept-incomplete task. We use the same splits and training augmentations as in CUB.

**AwA2**  The AwA2 task is constructed from the Animals with Attributes 2 (Xian et al., 2018) dataset. Each sample in this task consists of a normalised $(3, 224, 224)$ RGB image of an animal annotated with one out of 50 species classes (e.g., "zebra", "polar bear", etc.). In addition to a species label, each sample is annotated with 85 binary attributes (e.g., "black", "white", "stripes", "water", etc.). We use these binary attributes as concept labels and split them across 28 groups of semantically related concepts. Specifically, we group concepts across the following categories: { "colour", "fur pattern", "size", "limb shape", "tail", "teeth type", "horns", "claws", "tusks", "smelly", "transport mechanism", "speed", "strength", "muscle", "movement move", "active", "nocturnal", "hibernate", "agility", "diet", "feeding type", "general location", "biome", "fierceness", "smart", "social mode", "nest spot", "domestic"}. Similar to CUB, all samples with the same class in this dataset share the same concept profiles. The train-validation-test data splits are produced via a random 60%-20%-20% split, and samples are randomly cropped and flipped during training as in CUB.

**AwA2-Incomplete**  The AwA2-Incomplete is constructed from the AwA2 task by selecting, at random, 10% of AwA2's concepts to use as training annotations. This resulted in us selecting the following 9 concepts {"black", "gray",

"stripes", "hairless", "flippers", "paws", "plains", "fierce", "solitary"}. This dataset provides another example of a more realistic concept-annotated dataset where the concept labels are not complete descriptions of the downstream task.

**CIFAR10**  To explore our method in tasks without concept labels, we incorporate the `CIFAR10` (Krizhevsky et al., 2009) dataset as part of our evaluation. In this image detection task, each sample is a $(3, 32, 32)$ normalised RGB image that can be one out of 10 object types (e.g., "aeroplanes", "cars", "birds", "cats", etc.). As done by Marcinkevičs et al. (2024) and Vandenhirtz et al. (2024), we annotate all samples in this dataset with 143 textual concepts whose semantics were obtained in an unsupervised manner by Oikarinen et al. (2023). This allows us to construct numerical, unnormalised concept scores using the CLIP (Radford et al., 2021) similarity score between each image and each concept's textual description.

In contrast to how Marcinkevičs et al. (2024) and Vandenhirtz et al. (2024) binarise these concept scores, however, we do not use a zero-shot CLIP classifier selecting between a concept description and its textual negation. This is because generating binary concept labels in this manner leads to significantly imbalanced labels. Hence, such labels are extremely difficult for models to accurately learn as measured by their mean concept AUCs (although one can easily train models that achieve high concept accuracies, similar to those reported in (Marcinkevičs et al., 2024), given the high label imbalance). Therefore, here, we instead binarise concepts by thresholding their scores based on their 50th percentiles, as estimated from the entire dataset. This leads to concepts that are both balanced and still informative (e.g., we can see that our Bayes Classifier achieves high accuracies when provided with all concept labels in Figure 4 (bottom)). Nevertheless, note that by relying on an external, unsupervised model such as CLIP to annotate these labels, this dataset will undoubtedly contain noisy concept labels that do not always accurately represent their intended semantics.

**CelebA**  Our `CelebA` task is the same as used by Espinosa Zarlenga et al. (2022) based on the Large-scale CelebFaces Attributes dataset (Liu et al., 2018). Here, every sample is an annotated normalised RGB facial image that has been downsampled to have shape $(3, 64, 64)$. As in (Espinosa Zarlenga et al., 2022), we select the top-8 most balanced attributes from all of `CelebA`'s 40 attributes to construct a downstream label $y$ as the decimal representation of the vector containing all 8 attributes. We point out that although this process can yield up to 256 distinct task labels, in practice, we could only find 230 of them within the training samples, and we noticed that these labels were highly imbalanced (which results in a very difficult task to solve). To make this task concept-incomplete, we provide as concept annotations only the top-6 most balanced attributes, leaving two of the concepts needed to predict $y$ out of our training annotations. Finally, as in (Espinosa Zarlenga et al., 2022) and for consistency with previous works, our training set here is formed by randomly subsampling the original `CelebA`'s training set to a 12th of its size.

### TravelingBirds and TravelingBirds-Incomplete

The `TravelingBirds` task and its incomplete version are variations of their respective `CUB` tasks. These tasks, based on the `TravelingBirds` dataset proposed by Koh et al. (2020), introduce a new background to all images in `CUB`. This background is sampled from a category (e.g., "seashores", "forests", "coffee shops", etc.) which is, by design, correlated with the sample's task label (see Figure 8). As in the original `TravelingBirds` dataset, the test sets we use for these tasks exhibit a distribution shift where the background of each bird is assigned to a different category, making these tasks suitable test beds for generalisation and OOD evaluation.

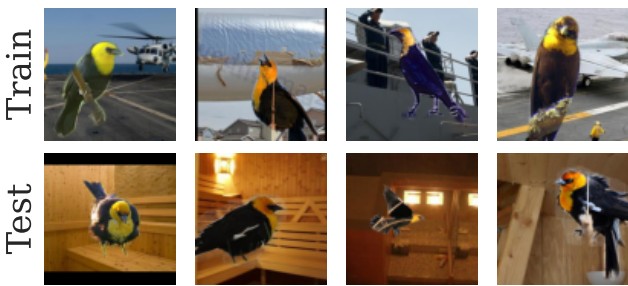

*Figure 8.* Randomly selected training and test samples of `TravelingBirds` for the class "Yellow-headed Blackbird". Notice that training samples all have "aircraft-carrier" backgrounds while the testing samples have "sauna" backgrounds.

# D. Training, Model Selection, and Hyperparameters

## D.1. Training

During training, we use the standard categorical cross-entropy loss as $\mathcal{L}_{\text{task}}$. For baselines that optimise a binary cross entropy loss between a predicted set of concepts and their corresponding ground truth labels, we use a weighted binary cross entropy loss that weights the loss of each concept's label proportionally to its representation in the training distribution (except in CelebA due to instabilities where we use an unweighted version). That way, we encourage models to learn useful concept predictors in tasks with high concept imbalance (e.g., CUB).

Unless specified otherwise, all baselines are trained using Stochastic Gradient Descent (SGD) with momentum $0.9$. When computing batch-level gradients, we use a batch size of $64$ for all CUB-based tasks (given their large sample and concept dimensions). Otherwise, we use a batch size of $512$ for all other tasks. Similarly, when possible, we fix the initial learning rate lr to values used by previous works and decay it during training by a factor of $10$ if the training loss reaches a plateau after 10 epochs. Specifically, we use $lr = 0.01$ for all tasks except for CelebA, where we use $lr = 0.05$ as in (Espinosa Zarlenga et al., 2022). Finally, based on the configuration by Koh et al. (2020), we use a weight decay $0.000004$ for the CUB-based and AwA2-based tasks.

All models were trained for a total of $E$ epochs, where $E = 150$ for all datasets except for CIFAR10, where it is $E = 50$. We use early stopping by tracking the validation loss and stopping training if an improvement in validation loss has not been seen after (patience) $\times$ (val_freq) epochs, where patience $= 5$ and val_freq, the frequency at which we evaluate our model on the validation set, is val_freq $= 5$.

## D.2. Base Architecture

Across all models, we use a ResNet-18 (He et al., 2016) pretrained on ImageNet as the backbone architecture $\psi$ for all tasks except for CelebA, where we use a larger model (a ResNet34) as the dataset is smaller. Specifically, we use the output of the second-to-last layer in ResNet (the layer before the original logits) as a backbone for all models. We do not freeze the initial pretrained weights. If the backbone is required to have a specific output dimension (e.g., as in Vanilla CBMs), then we achieve this by adding a leaky ReLU (Maas et al., 2013) nonlinearity and a linear layer with the correct output shape to the ResNet model mentioned above.

For the label predictor $f(\hat{\mathbf{c}})$, we use a single linear layer for all baselines. The only exception for this is ProbCBMs, where, as proposed by the original authors, we perform inference via a distance-based layer that learns class embeddings in $\mathbb{R}^{D_y}$ and compares their distance to a learnt linear projection of the bottleneck $\hat{\mathbf{c}}$ onto that space.

## D.3. Model Selection

Across all tasks and baselines, we performed a hyperparameter search with the aim of representing each baseline fairly. For this, we made significant efforts to provide each baseline with a similar fine-tuning budget. Moreover, we aimed to provide all baselines with the opportunity to have the same capacity as each other by including hyperparameterisations that lead to similar parameter sizes. When possible and available, we attempted to use the same hyperparameters and hyperparameter recommendations provided in the original works that proposed each baseline. Yet, it is worth noting that in some instances, our model selection process chose hyperparameters that resulted in models with fewer parameters, as they yielded better validation metrics.

After considering a model-specific selection of hyperparameters for each baseline (summarised in Table 3), we report the results of only the baseline with the highest area under its validation task accuracy vs intervention curve (the only exception being DNN, as it is uninterpretable and therefore we perform model selection based on its validation task). We use this area as a proxy for a metric that captures task fidelity, concept fidelity, and intervenability. Below, we provide details of all baselines, together with the hyperparameters that our model selection yielded for each of them across our tasks.

## D.4. Baseline Details and Selected Hyperparameters

**Vanilla CBM** All Vanilla CBM results in the main body of this work are produced from a jointly trained CBM (with concept weight loss $\lambda_c$) whose bottleneck is sigmoidal (i.e., $\hat{\mathbf{c}}$ is in $[0, 1]^k$). Our model selection here yielded $\lambda_c = 1$ for all concept-incomplete tasks CUB-Incomplete, AwA2-Incomplete, and CelebA. In contrast, for all other (concept complete) tasks, our model selection chose $\lambda_c = 10$.

*Table 3.* Sets of hyperparameters considered when performing our model selection. For each hyperparameter, we indicate which baseline(s) that hyperparameter is relevant to. For clarity, we separate MixCEM's specific hyperparameters at the bottom part of this table.

| Hyperparameter | Semantics | Searched Values | Baselines Fine-tuning These Hyperparameters |
|---|---|---|---|
| $\lambda_c$ | Concept loss weight | $\{1, 5, 10\}$ | CBM, Hybrid CBM, CEM, IntCEM |
| $k'$ | Extra unsupervised bottleneck dimensions | $\{0, 50, 100, 200\}$ | Hybrid CBM, DNN |
| $m$ | Concept embedding space dimension | $\{16, 32\}$ | CEM, ProbCBM |
| $D_y$ | Class embedding space dimension | $\{64, 128\}$ | ProbCBM |
| $\gamma$ | Training intervention loss penalty | $\{1.1, 1.5\}$ | IntCEM |
| $\lambda_{\text{roll}}$ | Intervention policy regulariser | $\{0.1, 1, 5\}$ | IntCEM |
| $\lambda_{\text{complex}}$ | Complexity regulariser | $\{0.000001, 0.001, 0.1\}$ | P-CBM, Residual P-CBM |
| $\lambda_p$ | Prior loss weight | $\{0.1, 1\}$ | MixCEM |
| $p_{\text{drop}}$ | Residual dropout probability | $\{0.1, 0.5, 0.9\}$ | MixCEM |
| $E_{\text{cal}}$ | Number of Platt scaling epochs | $\{0, 30\}$ | MixCEM |

**Hybrid CBM** Hybrid CBMs (Mahinpei et al., 2021) are variations of jointly trained sigmoidal CBMs where the concept bottleneck $\hat{\mathbf{c}} = [\hat{\mathbf{c}}_{\text{aligned}}, \hat{\mathbf{c}}_{\text{unaligned}}]^T \in \mathbb{R}^{(k+k')}$ is formed by the concatenation of a binary component $\hat{\mathbf{c}}_{\text{aligned}} \in [0,1]^k$, whose $i$-th entry is trained to be aligned with the $i$-th ground-truth concept, and a real-valued unconstrained component $\hat{\mathbf{c}}_{\text{extra}} \in \mathbb{R}^{k'}$, whose $k'$ entries are not aligned to any known concept. Our model selection for Hybrid CBMs yielded $k' = 50$ for all tasks except for CelebA, which yielded $k' = 200$. Similarly, we selected $\lambda_c = 10$ for CelebA, AwA2, and AwA2-Incomplete, $\lambda_c = 1$ for CUB-Incomplete and CIFAR10, and $\lambda_c = 5$ for CUB.

**CEM** When training CEMs, we intervene on a concept with probability $p_{\text{int}} = 0.25$ (as suggested by the authors (Espinosa Zarlenga et al., 2022)). In this setup, our model selection selected $m = 16$ for CEM's embedding size for all tasks except for CelebA, where we obtained $m = 32$. Finally, we used the following concept loss weights: $\lambda_c = 10$ for CelebA, $\lambda_c = 5$ for AwA-Incomplete, and $\lambda_c = 1$ for all other tasks.

**IntCEM** Given the number of hyperparameters in IntCEMs, we focused on fine-tuning only the concept loss weight $\lambda_c$, the training-time task intervention penalty $\gamma$, and the intervention policy regulariser $\lambda_{\text{roll}}$. Therefore, we fixed the embedding size $m$ to that used by the equivalent CEM in the respective task, the maximum number of training-time interventions $T$ to 6, the initial probability of intervention as $p_{\text{int}} = 0.25$, and the annealing rate for $T_{\text{max}}$ to 1.005. We chose these values based on the suggestions by the original authors of this work. This resulted in the following hyperparameters being selected: (a) $\lambda_c = 1$ for all tasks, (b) $\gamma = 1.5$ for CUB, CelebA, AwA2, and AwA-Incomplete and $\gamma = 1.1$ for all other tasks, and (c) $\lambda_{\text{roll}} = 0.1$ for AwA2-Incomplete, $\lambda_{\text{roll}} = 5$ for CIFAR10, and $\lambda_{\text{roll}} = 1$ for all other tasks. Finally, for stability, we use global gradient clipping (clipping value of 100) for CelebA.

**ProbCBM** We attempt to closely follow the same hyperparameters for ProbCBMs used in the original work by Kim et al. (2023). As such, we (1) always use an Adam (Kingma & Ba, 2014) optimiser, (2) use a starting learning rate of 0.001 (except for CIFAR10 where we increase it to 0.01 as otherwise the model severely underperformed), (3) fix the number of training and inference samples to 50, (4) intervene on concepts during training with probability $p_{\text{int}} = 0.5$, (5) warm-up the model for 5 epochs, and (6) scale the KL divergence regulariser $\lambda_{\text{KL}} = 1 \times 10^{-5}$, as these were the hyperparameters used on the authors' original experiments. Similarly, we use weight decay $1 \times 10^{-6}$, a learning rate 10 times smaller for the non-pretrained weights, and clip gradient norms to 2 as the authors do in their official code base[2].

As suggested by the authors, we trained ProbCBMs in a sequential manner. For this, we use early stopping for a maximum of $E$ epochs but, for fairness, we spend at most $E/2$ of those epochs training the concept encoder and the remaining epochs training the task predictor. This left us with fine-tuning the dimensionality of both the concept embeddings ($m$) and the class embeddings ($D_y$), two hyperparameters which we noticed had a critical role in ProbCBM's performance. Here we use (a) $m = 32$ for CUB and CelebA, and $m = 16$ otherwise, and (b) $D_y = 128$ for CUB and AwA2, and $D_y = 64$ otherwise.

As in (Kim et al., 2023), we perform interventions in ProbCBMs by replacing sampled concept embeddings with the learnt embedding means of their corresponding ground-truth labels.

**Posthoc CBM** We train both the standard and residual versions of P-CBMs. Here, we first train a DNN on the downstream task $y$ for $E$ epochs by attaching two linear layers, with a leaky ReLU between them, to the output of the backbone $\psi$. The first linear layer will have as many neurons as concepts in the dataset and the second layer will have as many neurons as

---

[2]See https://github.com/ejkim47/prob-cbm/blob/main/configs/config_exp.yaml#L30.

output labels in the specific task. This is done so that we provide this model with a similar capacity to the CBMs we train. We train the black-box model with the same optimiser's hyperparameters used for other baselines.

Once the black box task predictor has been trained, we extract a training set of embeddings by projecting the entire training set to the space of the second-to-last layer of this model (i.e., the space of the first linear layer with $k$ neurons that we added). Then, we learn the Concept Activation Vector (CAVs) (Kim et al., 2018) for concept $c_i$ using the vector perpendicular to the decision boundary of a linear SVM, with $\ell_2$ penalty $C = 1$, trained to predict concept $c_i$ from the activations of the second-to-last layer of the black box DNN.

When fine-tuning P-CBMs's sparse linear classifier, we fine-tune the complexity regulariser $\lambda_{\text{complex}}$ and fix the elastic net's $\ell_1$ ratio to be 0.1. Our model selection chose $\lambda_{\text{complex}} = 1 \times 10^{-6}$ for CUB, AwA2, and CIFAR10, $\lambda_{\text{complex}} = 0.001$ for CUB-Incomplete and AwA2-Incomplete, and $\lambda_{\text{complex}} = 1$ for all other tasks.

When training the residual version of P-CBM, we provided the residual layer with $k$ hidden neurons to enable this model to have a closer capacity than that of competing baselines. Our fine-tuning for this model selected $\lambda_{\text{complex}} = 1 \times 10^{-6}$ for CUB, $\lambda_{\text{complex}} = 0.001$ for CIFAR10, and $\lambda_{\text{complex}} = 0.1$ for all other tasks.

Finally, given a lack of a direct mechanism for performing concept interventions on P-CBMs, when we intervene on their concept predictions, we follow the same intervention process as in Vanilla CBMs, whose bottlenecks are unnormalised (e.g., logits). That is, as suggested by Koh et al. (2020), we indicate a concept $c_i$ is active by setting the neuron aligned to its score to the 95th percentile value of that neuron in the empirical training distribution. Similarly, we indicate a concept is inactive by setting that same neuron's output to the 5th percentile of its empirical training distribution.

**DNN** As a representative of black-box models, in our evaluation, we included vanilla Deep Neural Networks (DNNs). To ensure fairness in terms of capacity with respect to other baselines, we implemented DNNs using the same architecture as Hybrid CBMs in each task but setting the concept weight to 0. This ensures this model only learns its downstream task using the same architecture provided to equivalent Hybrid CBMs. Therefore, as in Hybrid CBMs, we fine-tuned the number of extra dimensions $k'$ in the bottleneck that follows the backbone, selecting a value in $k' \in \{0, 50, 100, 200\}$, and set the activation function of the entire bottleneck to a vanilla leaky ReLU function. Our model selection chose $k' = 200$ for CUB, CUB-Incomplete, AwA2, and CelebA while it chose $k' = 100$ for AwA2-Incomplete and CIFAR10.

**MixCEM** For MixCEMs, we fine-tune the weight of the prior loss $\lambda_p$, the dropout probability of a concept embedding's residual during training $p_{\text{drop}}$, and the number of epochs $E_{\text{cal}}$ used for Platt calibration (Platt et al., 1999) on the validation set. All other hyperparameters that are inherited from CEMs, such as the probability of training intervention $p_{\text{int}}$, the concept loss weight $\lambda_c$, and the embedding size $m$, are always set to the same values selected for CEMs on the respective task. As for our other hyperparameters, our model selection yielded: (a) $\lambda_p = 1$ for all tasks except for CIFAR10, where we selected $\lambda_p = 0.1$, (b) $p_{\text{drop}} = 0.1$ for CUB-Incomplete and the TravelingBirds-based tasks, $p_{\text{drop}} = 0.5$ for CUB, AwA2, and CIFAR10, and $p_{\text{drop}} = 0.9$ for CelebA, and (c) $E_{\text{cal}} = 30$ for all tasks except for CelebA, where model selection chose not to perform Platt Scaling (i.e., $E_{\text{cal}} = 0$). Finally, we always fix the number $M$ of residual dropout samples we generate at inference to $M = 50$.

In Appendix J, we show an ablation study for our model's hyperparameters that suggests its ability to achieve both CA and BI is preserved across several hyperparameterisations. Nevertheless, noticing that $(\lambda_p, p_{\text{drop}}, E_{\text{cal}}) = (1, 0.5, 30)$ were by far the most often selected hyperparameters, we recommend using these as the default values of MixCEM's hyperparameters if there are no resources for fine-tuning its hyperparameters.

### D.5. Bayes Classifier

To determine whether or not all baselines achieve bounded intervenability, we wish to approximate the accuracy of a Bayes Classifier that takes as input any set of ground-truth concept labels $\mathbf{c}_S$ and predicts $\arg\max_{l \in \{1, \cdots, L\}} \mathbb{P}(y = l \mid \mathbf{c}_S)$. As learning a model for each possible concept subset $S \subseteq \{1, \cdots, k\}$ is intractable, we approach this problem by approximating the Bayes Classifier via a Multilayer Perceptron (MLP) $\eta(\mathbf{c}) : [0, 1]^k \to [0, 1]^L$ and training it to minimise $\mathcal{L}_{\text{task}}(y, \eta(\mathbf{c}))$. For this model to support any arbitrary concept subset as an input during inference, however, we randomly mask its input concept vectors $\mathbf{c}$ during training by setting any input concept to 0.5 with probability $p = 0.25$. That way, given the labels $\mathbf{c}_S$ of any subset of concepts $S$, we can estimate $\mathbb{P}(y \mid \mathbf{c}_S)$ by predicting $\eta(\mathbf{c}'(S))$, where we let $\mathbf{c}'(S) \in [0, 1]^k$ be a vector such that $\mathbf{c}'(S)_i = \mathbf{c}_i$ if $i \in S$ and $\mathbf{c}'(S) = 0.5$, otherwise. In practice, across all tasks, we train this masked model for $E_{\text{Bayes}} = 75$ epochs using an MLP with hidden layers $[28, 64, 32]$ and leaky ReLU nonlinearity in between them.

# E. Results on Variations of Vanilla CBMs

As discussed in Section 2, Vanilla CBMs can be trained jointly, sequentially, and independently. Moreover, the jointly trained version of Vanilla CBMs can use a bounded sigmoidal activation for its bottleneck, or it can use unbounded logit scores. In the latter case, where the bottleneck contains the logits of the concept they predict, these scores can be intervened on using the 5th and 95th percentile values of their respective training distributions as suggested by Koh et al. (2020). Here, we investigate how various training regimes for Vanilla CBMs impact their ID and OOD intervention performances.

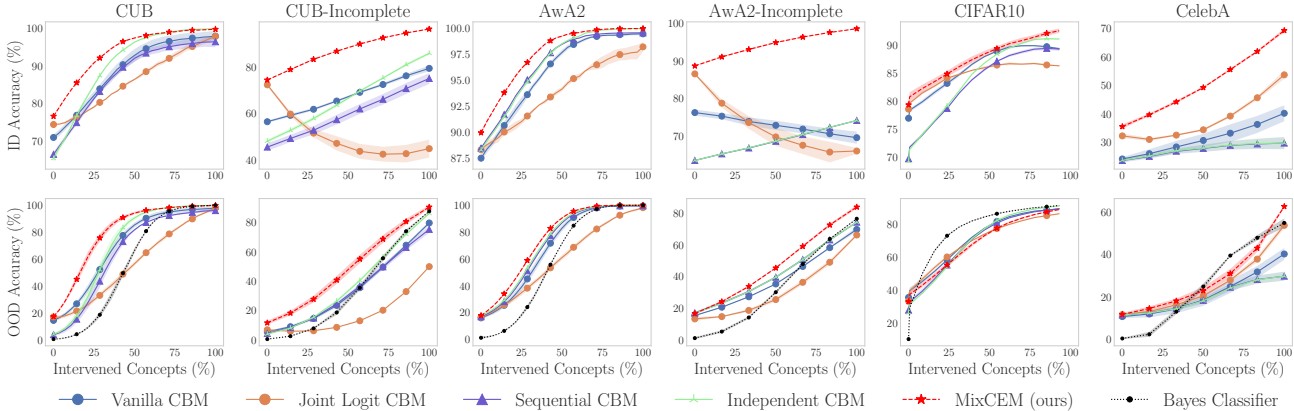

*Figure 9.* Task accuracy of several Vanilla CBM variants as we intervene on concepts, selected at random, for ID (top) and OOD (bottom) test samples. We use the same setup as the experiments described in Figure 4.

In Figure 9, we show the results of intervening on several variations of Vanilla CBMs across all tasks and compare these to our MixCEM's results. These results, which follow the same setup introduced for Figure 4 in Section 5, show that MixCEM outperforms all CBM variations when intervened on for ID and OOD samples. In particular, we observe that MixCEMs significantly outperform all variants of Vanilla CBMs in our concept-incomplete tasks as, in contrast to Vanilla CBMs, MixCEMs are completeness-agnostic. The closest variant to MixCEM in terms of its unintervened performance is the jointly trained logit CBMs, as their logits are leakage-enabling activations in the bottleneck. Yet, as seen in particular in the concept-incomplete tasks such as `CUB-Incomplete` and `AwA2-Incomplete`, these models are unable to properly react to interventions in these setups, instead decreasing their accuracy the more they are intervened on.

# F. Scalability Studies

Below, we describe a series of experiments to measure the computational impact of MixCEM's architectural components on MixCEM's training and inference times.

### F.1. Training Times

In Table 4, we show the efficiency of each method expressed as the number of wall-clock seconds taken per training epoch across all tasks. We emphasise that these results are likely biased as they depend on implementation. Moreover, they are prone to high variance due to our hardware infrastructure (we train models on shared machines whose latency may be affected by concurrent processes). Nevertheless, our results suggest that MixCEM's training times are significantly faster than those seen in IntCEMs and ProbCBMs. This is due to IntCEMs incorporating an expensive sampling-based training objective, which MixCEMs avoid completely. As expected, MixCEM's times are slower than CEMs given their introduction of new mechanisms on top of CEM. Nevertheless, we believe this performance hit is not significant and can be considered to be amortised at inference time if one considers that MixCEMs are much better at receiving concept interventions than CEMs both for ID and OOD samples. Finally, we observe that as the number of concepts increases, the training times of ProbCBM rise significantly, even for similar sample sizes (e.g., `AwA2` and `CIFAR10`). This is highly suggestive that these models may not properly scale to large concept bases and suggests that MixCEMs, whose performance remains relatively stable even when the number of concepts is high, can properly scale across concept set sizes.

*Table 4.* Efficiency study showing the training time per epoch (in seconds) for all embedding-based baselines.

| Method | CUB | CUB-Incomplete | AwA2 | AwA2-Incomplete | Cifar10 | CelebA |
|---|---|---|---|---|---|---|
| ProbCBM | $67.90_{\pm 6.54}$ | $26.08_{\pm 2.67}$ | $182.38_{\pm 6.09}$ | $121.07_{\pm 25.12}$ | $428.52_{\pm 156.10}$ | $11.14_{\pm 0.12}$ |
| CEM | $30.34_{\pm 2.87}$ | $26.21_{\pm 1.49}$ | $82.44_{\pm 9.38}$ | $108.82_{\pm 12.63}$ | $32.37_{\pm 0.52}$ | $8.15_{\pm 0.19}$ |
| IntCEM | $53.75_{\pm 8.81}$ | $35.57_{\pm 4.27}$ | $149.96_{\pm 8.94}$ | $144.18_{\pm 35.97}$ | $93.85_{\pm 1.58}$ | $11.57_{\pm 0.41}$ |
| MixCEM | $50.03_{\pm 3.31}$ | $26.00_{\pm 0.90}$ | $110.94_{\pm 1.27}$ | $97.96_{\pm 6.81}$ | $59.92_{\pm 2.76}$ | $7.73_{\pm 0.02}$ |

### F.2. Inference Times

To complement our discussion on MixCEM's training times with respect to competing baselines, in this section we explore the effect of MixCEM's components on its inference time. Table 5 shows the inference wall-clock times of all baselines in CUB. Although we observe a slight increase in inference times of MixCEMs with respect to CEMs ($\sim$4.2% slower), we argue this difference is not problematic, considering MixCEM is designed to be deployed together with an expert who can intervene on it. In this setup, we believe that less than a millisecond of extra latency should not bear too heavy a toll, as post-intervention accuracy is more important.

*Table 5.* Efficiency study showing the inference time per sample (in milliseconds) for all baselines in CUB.

| | Vanilla CBM | Hybrid CBM | ProbCBM | P-CBM | Residual P-CBM | CEM | IntCEM | MixCEM |
|---|---|---|---|---|---|---|---|---|
| Time per Sample (ms) | 1.386 | 1.394 | 5.687 | 1.426 | 1.433 | 1.437 | 1.431 | 1.497 |

## G. Image Noising Details

Across all of our experiments where noise is used to generate OOD test samples (e.g., Figure 4 and Figure 5, left), we use a form of "Salt & Pepper" noise. Given a noise strength factor $\lambda_l \in [0, 1]$, this noise sets a randomly selected fraction of $\lambda_l/2$ pixel channels, selected with replacement for efficiency, to 255, their maximum value. Then, it sets $\lambda_l/2$ randomly selected pixel channels, selected with replacement for efficiency, to 0, their minimum value. This leads to the resulting image having *at most* $\lambda_l$ of its channels corrupted and to them becoming darker as the noise level increases. If the noise level is not specified, then we use $\lambda_l = 10\%$ as the default level. We note that we use this version of Salt & Pepper noise as it allows us to efficiently test its effects on models across large datasets using noise that is similar to that found in real-world scenarios (Azzeh et al., 2018). A visualisation of CUB images with different levels can be seen in Figure 10.

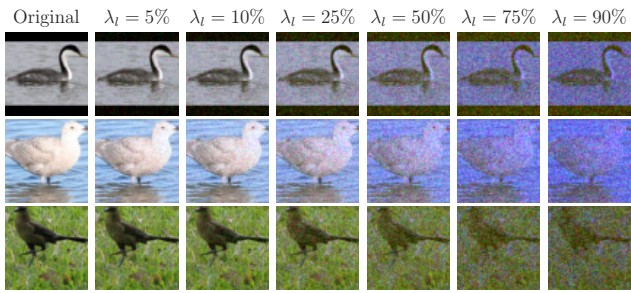

*Figure 10.* Examples of random images in CUB with our form of "Salt & Pepper" noise as we vary the noise's strength level.

## H. Interventions on Different Forms of Distribution Shifts

In this section, we explore different forms of distribution shifts beyond those studied in Section 5. Our results below strongly suggest that the improvements in intervenability described in Section 5 for MixCEMs can be seen across multiple forms of distribution shifts.

## H.1. Exploring Different Forms of Visual Distribution Shifts

First, we explore visual distribution shifts caused by different forms of transformations besides the Salt & Pepper noise we studied in Section 5.2. Specifically, in this section, we evaluate interventions on samples that were downsampled, Gaussian-blurred, and subjected to a random affine transformation (including rescaling and rotation). We chose these distribution shifts as they represent widespread forms of OOD shifts found in real-world deployment.

Our results on `CUB-Incomplete` and our `AwA2` tasks, shown in Figure 11, suggest that MixCEMs have better OOD intervention task accuracies than our baselines across different distribution shifts. For instance, MixCEMs can have up to ∼20% percentage points more in OOD task accuracy than CEMs and IntCEMs when all concepts are intervened on in inputs downsampled to 25% of their size. These results suggest that MixCEMs are better at receiving interventions in practical scenarios and real-world forms of distribution shifts.

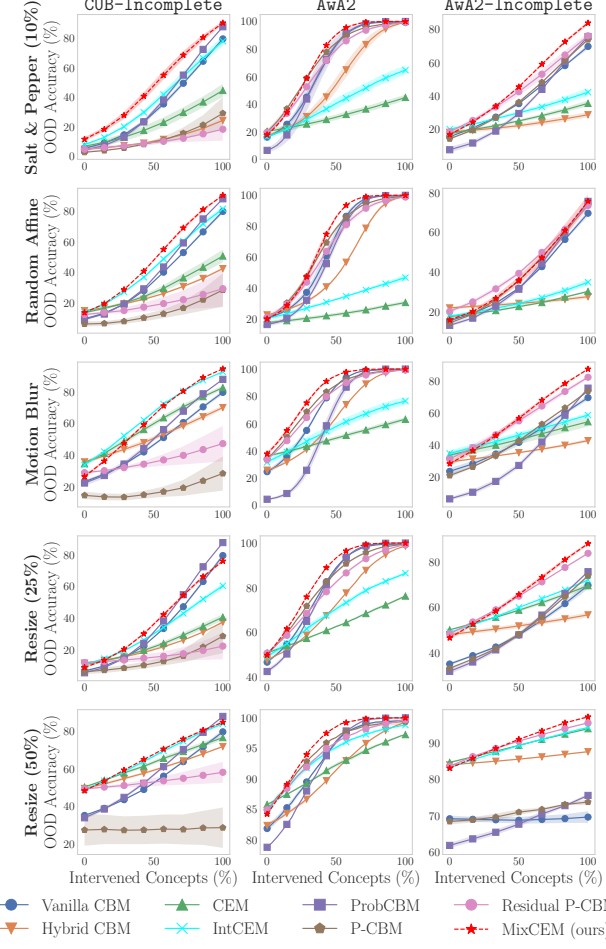

*Figure 11.* Task accuracy as we intervene on concepts, selected at random, for OOD test samples on `CUB-Incomplete` and our `AwA2` variants. Out-of-distribution samples are generated by applying different forms of visual transformations to the test set (shift type shown on the y-axis).

## H.2. Exploring Domain Shifts

Next, we explore distribution shifts in the form of domain shifts. For this, we train our models on an addition task where 11 `MNIST` digits (Deng, 2012) form each training sample, and the task is to predict whether all digits add to more than 25% of the maximum sum. We provide the identity of five digits as training concepts (i.e., it is an incomplete task), and at test time, we swap `MNIST` digits for real-world sampled digits coming from the Street View House Numbers (`SVHN`) dataset (Netzer et al., 2011). Our results, shown in Figure 12, suggest that MixCEMs achieve better ID and OOD intervention

task AUC-ROC than our baselines, particularly for high intervention rates. For example, when all concepts are intervened, MixCEM attained ∼31, ∼7, and ∼3 more percentage points in OOD task AUC-ROC over CEM, IntCEM, and ProbCBM, respectively. In contrast, we found it very difficult to get CEMs to perform well in this incomplete task.

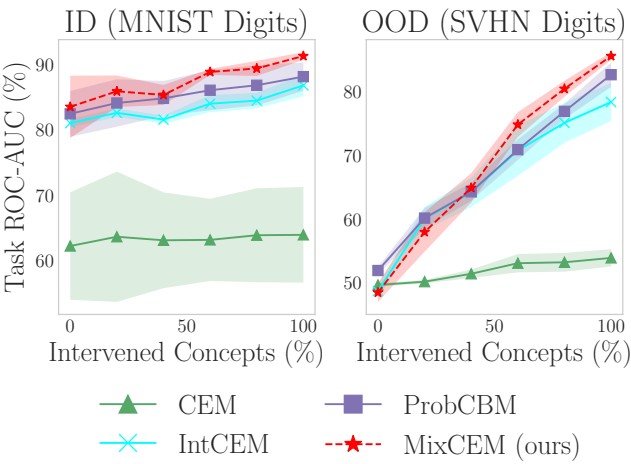

*Figure 12.* Task accuracy as we intervene on concepts, selected at random, for models trained on the digit addition task. Models are trained using `MNIST` digits. However, when we evaluate OOD interventions, we use test samples whose digits are drawn from the `SVHN` dataset.

# I. Extended Robustness Experiments

We complement our results in Section 5.3 by showing extended versions of those experiments. We first discuss our extended noise level ablation results on `CUB-Incomplete` and `CUB` and then our baselines' results on `TravelingBirds`.

## I.1. Extended Noise Ablation Results

Figure 13 shows all intervention curves in `CUB` and `CUB-Incomplete` as we vary the amount of "Salt & Pepper" noise on the test samples. As in our analysis of Figure 5 (left) in Section 5.3 suggests, we see that across noise levels, MixCEM outperforms all baselines in terms of its task accuracy as interventions are made. MixCEM's placement within the other baselines appears to be independent of the dataset's concept completeness, although we observe a significantly greater improvement in `CUB-Incomplete` than in `CUB`. Moreover, we also observe that, as discussed in Section 5.3, MixCEM's unintervened performance (left-most part of the plot) is above that of other methods for noise levels of up to 10% corruption. After those levels, MixCEM's unintervened accuracy seems to be on par with that of all other baselines, but it better receives interventions than other approaches (leading to higher accuracies when intervened on across all noise levels).

## I.2. Complete Spurious Study on `TravelingBirds`

In Figure 14, we summarise the intervention curves for all baselines in `TravelingBirds` and its incomplete version `TravelingBirds-Incomplete`. Here, as we observed in Section 5.3, we see that MixCEM's performance for spuriously correlated ID samples is on par with that of IntCEMs and CEMs, the two best-performing baselines for ID samples in both tasks. In contrast, for OOD samples, we see that ProbCBMs, Vanilla CBMs, and Hybrid CBMs have better OOD unintervened performance than MixCEM only in `TravelingBirds`. This aligns with the results we reported in Section 5.3 and constitutes evidence for the hypothesis we discuss in Section 6 that argues that using constant/global concepts in a bottleneck, such as those used for Vanilla CBMs and ProbCBMs, enables them to avoid exploiting spurious correlations in the same way less constrained methods such as CEMs or IntCEMs may. These results also follow similar conclusions previously discussed in this dataset by Koh et al. (2020). Nevertheless, we observe that (1) MixCEM's accuracy is higher than that of all baselines when the dataset is incomplete, our main setup of interest, and (2) MixCEMs intervention accuracy is properly bounded (always above that of the Bayes Classifier), something that cannot be said of any other method in these two tasks except for IntCEMs.

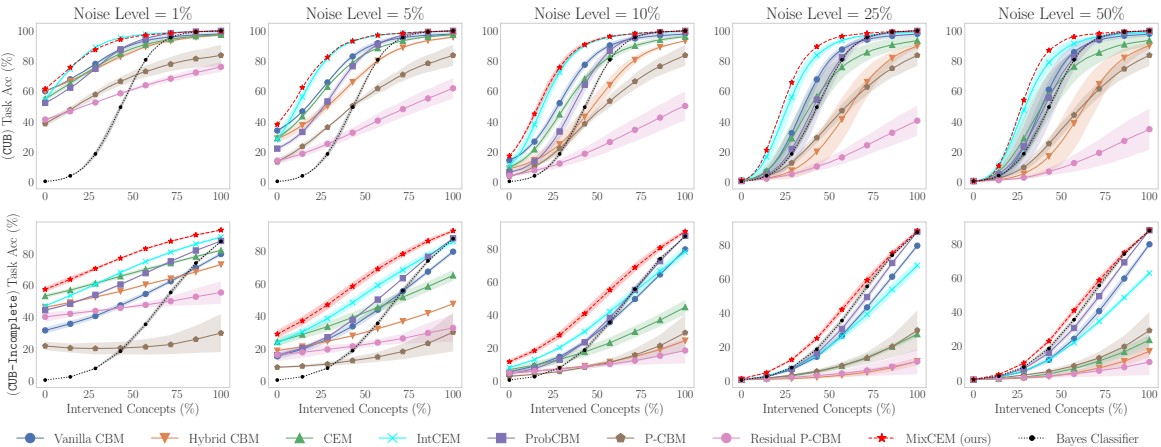

*Figure 13.* OOD task accuracy vs intervention curves for all baselines in the `CUB` (top) and `CUB-Incomplete` tasks. Test samples are perturbed by adding "Salt & Pepper" noise with increasing levels (% pixels corrupted).

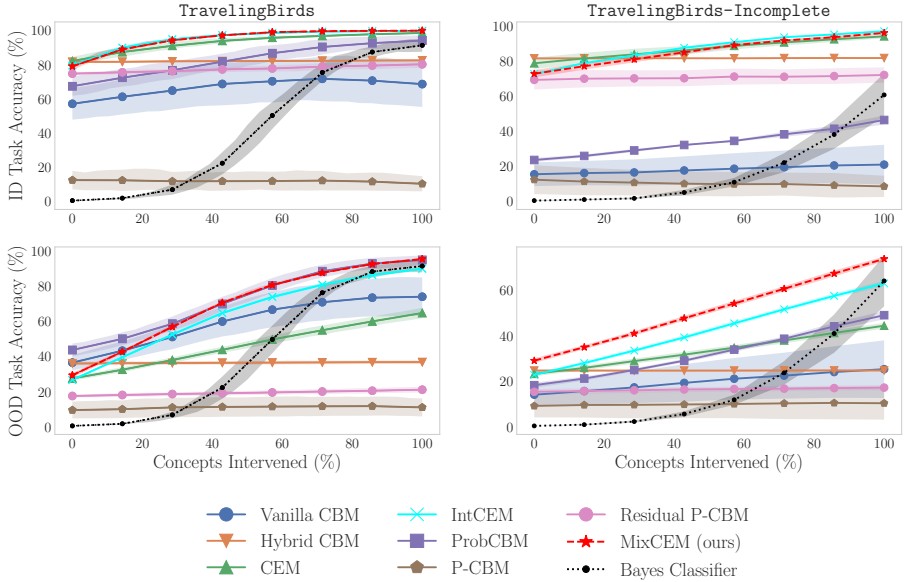

*Figure 14.* `TravelingBirds` (left) and `TravelingBirds-Incomplete` (right) intervention curves. We show results on a spuriously correlated ID validation set (top) and on an OOD test set without the spurious correlation (bottom).

## J. Hyperparameter Recommendations and Ablation Studies

In this section, we thoroughly examine the impact of MixCEM's hyperparameters on its performance. For this, we focus on the `CUB-Incomplete` dataset, given that it is a good representative for a concept-incomplete task, the main setup of interest for this paper. Moreover, to enable a tractable exploration of a vast number of configurations for the ablations below, we train MixCEM only with 25% of the training data. The only exceptions for this are in our studies of the effect of (1) random train-time interventions (Appendix J.5), and (2) concept calibration (Appendix J.4), where we conducted experiments across all tasks, as these studies could be performed efficiently. Finally, when studying a specific hyperparameter, we vary its value over a set of pre-defined acceptable values at different scales while fixing all other hyperparameters to the values selected by our model selection procedure for `CUB-Incomplete` (described in detail in Appendix D).

Our results in this section strongly suggest that **MixCEMs are robust to different hyperparameterisations**, attaining both high intervenability and fidelities for both ID and OOD samples across all hyperparameterisations we attempted. As

such, MixCEMs do not require significant efforts to fine-tune, making them practical in real-world setups where proper fine-tuning may be intractable. To facilitate MixCEM's future use, we include a **set of hyperparameter recommendations** in Appendix J.6.

### J.1. Effect of Concept Weight Loss ($\lambda_c$)

First, we evaluate the effect of the concept loss weight $\lambda_c$ on MixCEM's performance. Although, in practice, we do not fine-tune this hyperparameter for MixCEM (instead always using the concept loss weight selected for an equivalent CEM), this hyperparameter has been previously shown to significantly affect how CBM-based models perform (Koh et al., 2020; Espinosa Zarlenga et al., 2022). As such, understanding how it affects our model is an important practical consideration.

In Figure 15, we show the ID and OOD intervention curves for MixCEM in `CUB-Incomplete` across several values of $\lambda_c \in \{0, 0.01, 0.1, 1, 2.5, 5\}$. We see that, throughout all values of $\lambda_c$, MixCEM significantly increases its accuracy when one intervenes on its concepts both in ID and OOD test sets. Although these results appear to be somewhat stable, we see, as one would expect, that the unintervened task loss suffers a drop when $\lambda_c$ is high and the mean concept AUC drops when $\lambda_c$ is too low. This suggests that, in practice, using a midrange value for $\lambda_c$ in between $[1, 2.5]$ yields an intervenable model with high concept and task fidelities.

A surprising result from this ablation is that **MixCEM is still highly intervenable when $\lambda_c = 0$** (meaning no concept loss is applied during training). We believe that this is a consequence of MixCEM's prior predictive error $\mathcal{L}_{\text{task}}\big(y, f(\bar{\mathbf{c}})\big)$ maximising the task predictor accuracies assuming all concepts are intervened on. Hence, these results provide further evidence that $\mathcal{L}_{\text{task}}\big(y, f(\bar{\mathbf{c}})\big)$ has an implicit intervention-aware effect.

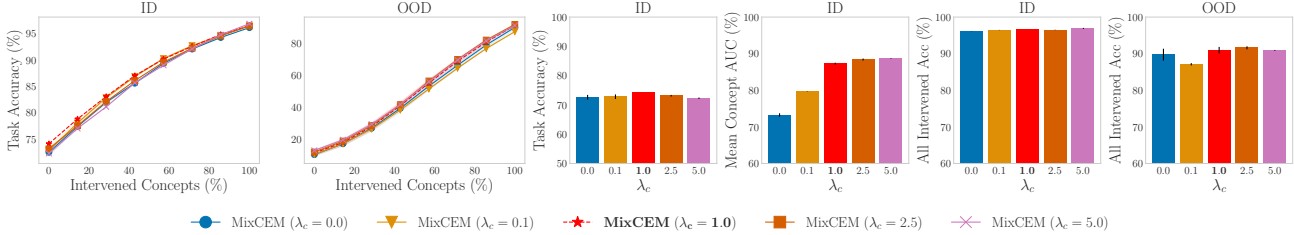

*Figure 15.* Ablation study for $\lambda_c$ in a smaller version of `CUB-Incomplete`. On top of each subplot, we indicate whether we show results for an ID test set or an OOD test set (generated using 10% salt & pepper noise). The right-most two plots show the task accuracy when all concepts are intervened. We highlight the model corresponding to the hyperparameter selected for our evaluation in Section 5.

### J.2. Effect of Prior Error Weight ($\lambda_p$)

Next, we examine how the prior error weight $\lambda_p$ affects MixCEM's performance. In Figure 16, we observe that small values of $\lambda_p$ result in both worse ID and OOD intervenability. This is expected, as the larger $\lambda_p$ is, the better the model's accuracy will be when making predictions based only on the global embeddings. Nevertheless, we see that for larger values of $\lambda_p$ (e.g., $\lambda_p \geq 1$), all metrics become relatively stable. As such, it is important that $\lambda_p$ is set to a value near 1. By doing so, during training we are assigning equal weight to correctly predicting task labels using *only* the global embeddings (i.e., minimising $\mathcal{L}_{\text{task}}\big(y, f(\bar{\mathbf{c}})\big)$) and to correctly predicting task labels with the contextual embeddings (i.e., $\mathcal{L}_{\text{task}}\big(y, f(g(\mathbf{x}))\big)$).

### J.3. Effect of Training Fallback Probability ($p_{\text{drop}}$)

When looking at the effect of the dropout probability $p_{\text{drop}}$, Figure 17 suggest that, at least in `CUB-Incomplete`, this hyperparameter does not have much of an effect. This is true except for $p_{\text{drop}} = 1$, where we see a drop in unintervened task accuracy. This is because when $p_{\text{drop}} = 1$, the model essentially blocks any leaked information from $\mathbf{x}$ into the label predictor $f$. However, for all other values, MixCEM's performance remains relatively stable with $p_{\text{drop}} = 0.5$, yielding the best overall intervenability results, albeit with a slight difference.

Surprisingly, however, even when $p_{\text{drop}} = 0$, meaning we do not use any residual dropout during training or testing, MixCEM achieves very high task accuracies when it is intervened on for ID and OOD samples. This may suggest that this dropout mechanism is not always needed. Nevertheless, as our results in Figure 18 comparing a MixCEM with dropout and a MixCEM without dropout in `CelebA` show, there are clear benefits of adding this dropout mechanism, particularly for

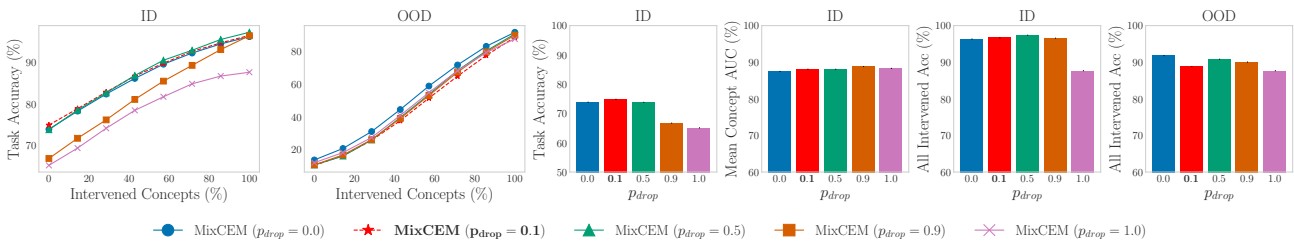

*Figure 16.* Ablation study for $\lambda_p$ in a smaller version of CUB-Incomplete. On top of each subplot, we indicate whether we show results for an ID test set or an OOD test set (generated using 10% salt & pepper noise). The right-most two plots show the task accuracy when all concepts are intervened. We highlight the model corresponding to the hyperparameter selected for our evaluation in Section 5.

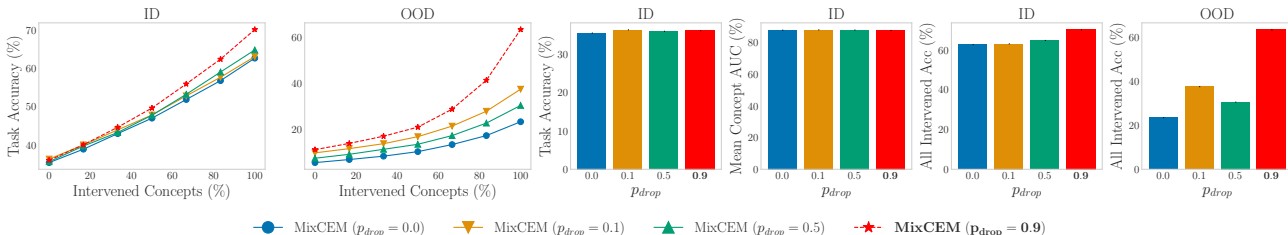

*Figure 17.* Ablation study for $p_{\text{drop}}$ in a smaller version of CUB-Incomplete. On top of each subplot, we indicate whether we show results for an ID test set or an OOD test set (generated using 10% salt & pepper noise). The right-most two plots show the task accuracy when all concepts are intervened. We highlight the model corresponding to the hyperparameter selected for our evaluation in Section 5.

difficult tasks such as CelebA.

*Figure 18.* Ablation study for MixCEM's residual dropout probability $p_{\text{drop}}$ hyperparameter in CelebA. Notice that in this task, there is a significant improvement in OOD intervenability when $p_{\text{drop}}$ is greater than 0. We mark in red and bold the model corresponding to the hyperparameter that was selected for our evaluation in Section 5.

### J.4. Effect of Calibration ($E_{\text{cal}}$)

In Figure 19, we can see the intervention curves of MixCEMs across all tasks with and without Platt scaling. We notice that Platt scaling brings some key benefits for OOD samples, particularly for complex datasets such as CUB-Incomplete and AwA2-Incomplete. This is because the more calibrated a MixCEM's concept probabilities are, the more likely it is to drop its residual embedding when that concept becomes OOD. The only instance where we saw a drop in performance when using Platt Scaling was in CelebA. We believe this is due to the concepts in this task being too complex/subjective to be properly predicted in the first place, leading to a model whose concept predictions were not overconfident even before Platt scaling was done. Nevertheless, we notice that *with and without Platt Scaling*, MixCEMs can recover very high accuracies when intervened for OOD setups. Hence, Platt scaling is helpful but not entirely necessary for MixCEM's ability to receive interventions for OOD samples properly. Because of this, when selecting hyperparameters for MixCEMs, we include $E_{\text{cal}} = 0$ (i.e., no calibration at all) as a hyperparameter option.

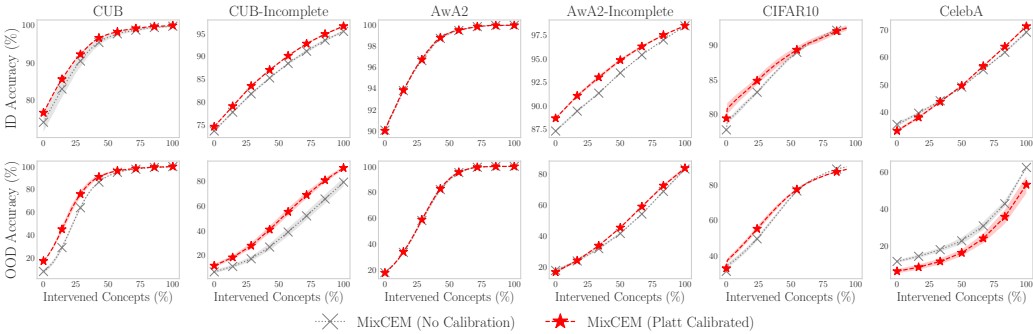

*Figure 19.* Task accuracy of MixCEMs with and without Platt scaling as we intervene on concepts, selected at random, for ID (top) and OOD (bottom) test samples. We use the same setup as the experiments described in Figure 4.

### J.5. Effect of Training Intervention Probability ($p_{\text{int}}$)

Finally, we study the effect of training-time interventions on MixCEM's performance across ID and OOD tasks (so-called *RandInt* (Espinosa Zarlenga et al., 2022)). Our results, shown in Figure 20 suggest that randomly intervening on concepts during training with $p_{\text{int}} = 0.25$ is generally beneficial for ID interventions. Nevertheless, we observe that the improvements from including these train-time interventions on MixCEMs are significantly less impactful than what the original CEM authors observed for CEMs (Figure 6 of (Espinosa Zarlenga et al., 2022)). This is once more evidence, as discussed in Section 6, that MixCEM's prior error minimisation has an implicit intervention-aware bias in it that leads to models that are more receptive to test-time interventions.

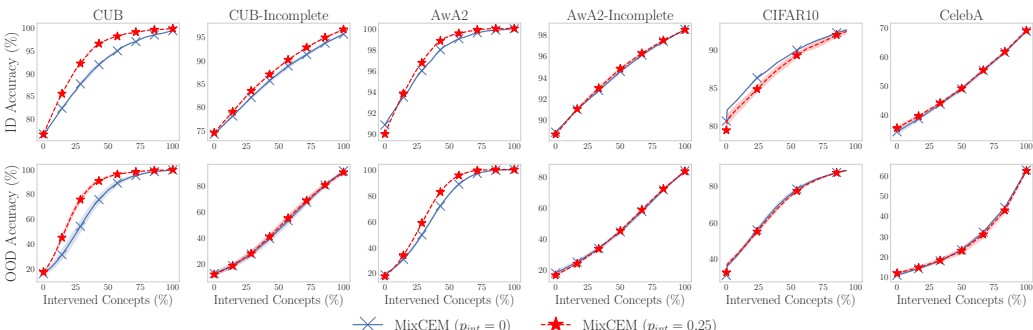

*Figure 20.* Task accuracy of MixCEMs with and without randomly intervening at training time with probability $p_{\text{int}}$ (i.e., using *RandInt*). We use the same setup as the experiments described in Figure 4.

### J.6. General Recommendations

Our ablation results suggest that MixCEMs can perform well both for ID and OOD instances across a wide range of hyperparameters. Thus, if fine-tuning is not an option, we would recommend setting $\lambda_c = 1$, $p_{\text{drop}} = 0.5$, $\lambda_p = 1$, $m = 16$, $p_{\text{int}} = 0.25$, and $T = 50$ for obtaining already high ID and OOD intervention receptiveness. If one of these hyperparameters is to be fine-tuned, our ablations suggest setting $\lambda_p = 1$ and focusing on fine-tuning $\lambda_c$, as changes in this hyperparameter affect MixCEM's interpretability the most.

We note that in our experiments, we focus almost entirely on selecting $\lambda_p$ and $p_{\text{drop}}$. All other hyperparameters (e.g., $m$, $p_{\text{int}}$, $T$) were either fixed to a constant value or were selected based on those used for an equivalent CEM.

## K. Learning Intervention Policies In MixCEM

As discussed in Section 6, because MixCEM utilises embedding representations for its concepts, we can train MixCEMs using the same intervention-aware pipeline used for IntCEMs in (Espinosa Zarlenga et al., 2023a). This would enable

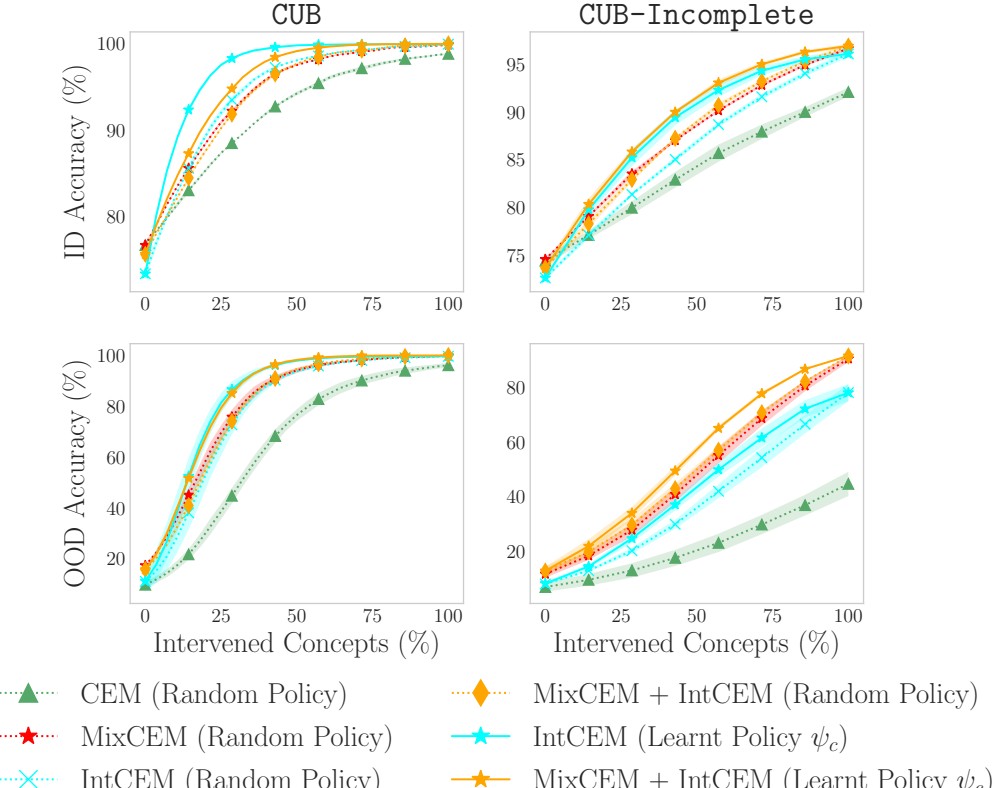

*Figure 21.* **Effect of integrating IntCEM's training procedure into MixCEM**. Intervention curves for ID (top) and OOD (bottom) test sets across our CUB-based tasks for learnt policies across intervention-aware models. As a baseline, we include intervention curves for CEM and MixCEM using a random policy. Otherwise, we show the curves produced from IntCEM-style learn policies $\psi_c$ using solid lines with stars. *MixIntCEM* corresponds to a MixCEM trained with IntCEM's training-time sampling mechanism, enabling the model to learn an intervention policy $\psi_c$ as a by-product.

MixCEM to learn an intervention policy during training, which can then be used at test time to express a preference over interventions.

In this section, we investigate the impact of incorporating MixCEM's training procedure, which involves sampling interventions at training time from a learnable policy $\psi_c$, into MixCEM's training objective. We note that, due to computational constraints, in this evaluation we did not perform a hyperparameter search over hyperparameters introduced when using IntCEM's training pipeline for MixCEM (e.g., $\lambda_{\mathrm{roll}}$ and $\gamma$) or over any of MixCEM's hyperparameters (e.g., $\lambda_c$, $\lambda_p$). Instead, we fix all hyperparameters of this hybrid model, which combines MixCEM and IntCEM, to those used for the equivalent MixCEM or IntCEM in the evaluation task. This may lead to results that are skewed against MixIntCEMs, as they were not explicitly fine-tuned. We summarise the results of our best-performing baselines in the CUB-based tasks, and show intervention curves when intervening following (1) a Random Policy and (2) the corresponding model's learnt policy $\psi_c$ (if applicable).

Our results in Figure 21 suggest that incorporating IntCEM's training procedure into MixCEM (a model we call *MixIntCEM*) does indeed lead to a model that learns an intervention policy $\psi_c$ that is significantly better than a Random Policy (as seen when looking at the differences between MixIntCEM's curves for the "Random Policy" vs the "Learnt $\psi_c$ Policy"). More importantly, we see that this benefit comes without sacrificing random intervenability in MixCEM, both for ID and OOD test sets (as seen by the almost identical random intervention curves for MixCEM and MixIntCEM). More importantly, however, we see that MixIntCEM's intervention policy yields better intervention curves than all other baselines for OOD test sets.

Nonetheless, we also observe that, for some instances, intervening on IntCEM using its learnt policy yields better intervention curves than those seen when we intervene on MixIntCEM using its own learnt policy. This is seen only when intervening on ID test sets for complete datasets (i.e., CUB), suggesting there is still some value in maintaining the original IntCEM pipeline if we know we are operating in a complete task where test samples will not go OOD. Considered together, these results suggest that, although MixCEM can learn a powerful intervention policy that works both for ID and OOD test sets, there are still benefits in using the "pure" IntCEM model in deployment if we know the model will be trained on a complete training set and evaluated uniquely on ID test sets.

## L. Resources Used

**Software** For our experiments and evaluation, we adapted the original CEM codebase[3] built by Espinosa Zarlenga et al. (2022). This codebase provided the basis for our implementation of MixCEM and gave us the foundations for our implementations of CEMs, IntCEMs, Vanilla CBMs and Hybrid CBMs. Moreover, we used the data loaders for CUB and CelebA provided by this codebase, of which the former is based upon the original data loader by (Koh et al., 2020). For Posthoc CBMs (P-CBMs), we based our implementation on that used by the authors and published with the paper[4]. Similarly, for Probabilistic CBMs, we based our implementation on a very close adaptation of the implementation made public by the authors in the codebase accompanying their paper[5]. Finally, for our CIFAR10 and AwA2 loaders, we got inspiration from the public implementation of these loaders by Vandenhirtz et al. (2024) and Marcinkevičs et al. (2024), respectively.

Our experiments were run on PyTorch 1.11.0 (Paszke et al., 2019) and facilitated by PyTorch Lightning 1.9.5 (Falcon, 2019). For our plots, we used matplotlib 3.5.1 (Hunter, 2007) and the open-sourced distribution of draw.io.

All the software and datasets used to build our code and to run our experiments were made available to the public via open-source licenses (e.g., MIT, BSD). To facilitate and encourage the reproduction of our results, we include all of our code, including configuration files to reproduce each of our experiments, in our supplementary submission for this work. **All of our code**, including configs and scripts to recreate results shown in this paper, can be found in CEM's official public repository found at https://github.com/mateoespinosa/cem.

**Resources** We executed all experiments on a shared GPU cluster with four Nvidia Titan Xp GPUs and 40 Intel(R) Xeon(R) E5-2630 v4 CPUs (at 2.20GHz) with 125GB of RAM. All of our experiments, including early development experiments, expensive ablation studies, and fine-tuning for all baselines, took approximately 500 compute hours.

## M. Tabulation of Intervention Results and Computation of Area Under the Intervention Curves

In Table 6, we show a tabulation of Figure 4 using a representative subset of the points shown in that figure. We also include an estimation of the area under each intervention curve (computed using a Riemann sum), which shows how MixCEM's intervenability is significantly better than that of competing methods, particularly for OOD setups.

---

[3]https://github.com/mateoespinosa/cem.
[4]https://github.com/mertyg/post-hoc-cbm.
[5]https://github.com/ejkim47/prob-cbm.

*Table 6.* Task accuracy (%) for ID (blue) and OOD (red) samples as we intervene on a larger fraction of randomly selected concept groups. These results show the same data as Figure 4 but in a more accessible form. Each task's best result, and those not significantly different from it (paired $t$-test, $p = 0.05$), are in **bold**. We note that MixCEM's unintervened accuracies (i.e., $0\%$) may differ very slightly from those in Table 1 as its inference is non-deterministic (see Section 4). We show the estimated area under each intervention curve in the "AUC" column.

| | Method | 0% | 20% | 40% | 60% | 80% | 100% | AUC |
|---|---|---|---|---|---|---|---|---|
| **CUB** | Vanilla CBM | $70.97_{\pm0.76}$ / $14.43_{\pm1.16}$ | $80.43_{\pm1.07}$ / $38.26_{\pm2.93}$ | $90.36_{\pm1.56}$ / $77.55_{\pm2.29}$ | $95.36_{\pm1.63}$ / $91.85_{\pm1.92}$ | $97.27_{\pm1.49}$ / $96.56_{\pm1.65}$ | $97.98_{\pm1.39}$ / $97.98_{\pm1.39}$ | $89.75_{\pm1.31}$ / $72.25_{\pm1.44}$ |
| | Hybrid CBM | $73.65_{\pm0.23}$ / $8.96_{\pm2.63}$ | $80.68_{\pm0.33}$ / $18.75_{\pm4.96}$ | $88.03_{\pm0.25}$ / $43.20_{\pm6.50}$ | $93.40_{\pm0.35}$ / $69.12_{\pm4.59}$ | $96.86_{\pm0.37}$ / $87.64_{\pm1.33}$ | $98.07_{\pm0.20}$ / $93.29_{\pm0.21}$ | $89.08_{\pm0.22}$ / $54.06_{\pm3.74}$ |
| | ProbCBM | $68.16_{\pm1.44}$ / $6.44_{\pm0.97}$ | $79.38_{\pm0.94}$ / $21.65_{\pm3.62}$ | $91.63_{\pm0.33}$ / $64.35_{\pm5.06}$ | $97.11_{\pm0.21}$ / $90.11_{\pm1.86}$ | $99.50_{\pm0.03}$ / $98.59_{\pm0.44}$ | $100.00_{\pm0.00}$ / $100.00_{\pm0.00}$ | $90.53_{\pm0.39}$ / $65.52_{\pm2.32}$ |
| | P-CBM | $69.00_{\pm0.69}$ / $3.76_{\pm0.72}$ | $77.00_{\pm1.19}$ / $15.45_{\pm1.60}$ | $82.35_{\pm1.82}$ / $38.51_{\pm2.43}$ | $84.41_{\pm3.22}$ / $56.92_{\pm4.36}$ | $84.82_{\pm4.81}$ / $74.47_{\pm5.72}$ | $83.51_{\pm7.01}$ / $83.81_{\pm6.94}$ | $81.15_{\pm2.90}$ / $45.83_{\pm3.00}$ |
| | Residual P-CBM | $71.84_{\pm0.57}$ / $4.26_{\pm0.86}$ | $76.32_{\pm0.63}$ / $9.75_{\pm2.06}$ | $80.59_{\pm0.70}$ / $18.73_{\pm4.34}$ | $83.44_{\pm0.96}$ / $28.55_{\pm6.60}$ | $86.31_{\pm1.07}$ / $40.56_{\pm8.58}$ | $88.13_{\pm1.26}$ / $50.30_{\pm9.66}$ | $81.40_{\pm0.75}$ / $25.01_{\pm5.16}$ |
| | Bayes Classifier | $0.52_{\pm0.00}$ / $0.52_{\pm0.00}$ | $9.31_{\pm0.87}$ / $9.31_{\pm0.87}$ | $49.51_{\pm2.20}$ / $49.51_{\pm2.20}$ | $86.40_{\pm0.62}$ / $86.40_{\pm0.62}$ | $98.68_{\pm0.11}$ / $98.68_{\pm0.11}$ | $100.00_{\pm0.00}$ / $100.00_{\pm0.00}$ | $58.62_{\pm0.74}$ / $58.62_{\pm0.74}$ |
| | CEM | $76.67_{\pm0.11}$ / $9.68_{\pm1.20}$ | $85.95_{\pm0.18}$ / $32.58_{\pm3.12}$ | $92.69_{\pm0.15}$ / $68.27_{\pm2.68}$ | $96.03_{\pm0.21}$ / $85.07_{\pm2.30}$ | $97.93_{\pm0.17}$ / $93.31_{\pm1.43}$ | $98.78_{\pm0.10}$ / $96.14_{\pm0.90}$ | $92.25_{\pm0.09}$ / $66.53_{\pm1.96}$ |
| | IntCEM | $73.33_{\pm0.70}$ / $11.00_{\pm4.01}$ | $90.23_{\pm0.38}$ / $57.05_{\pm5.26}$ | $97.22_{\pm0.14}$ / $90.27_{\pm1.16}$ | $98.79_{\pm0.07}$ / $96.54_{\pm0.31}$ | $99.61_{\pm0.02}$ / $98.92_{\pm0.10}$ | $99.90_{\pm0.01}$ / $99.67_{\pm0.12}$ | $94.96_{\pm0.08}$ / $80.19_{\pm1.97}$ |
| | MixCEM (ours) | $76.64_{\pm0.22}$ / $17.25_{\pm0.39}$ | $89.19_{\pm0.17}$ / $62.24_{\pm3.43}$ | $96.52_{\pm0.13}$ / $90.84_{\pm1.32}$ | $98.45_{\pm0.07}$ / $96.78_{\pm0.58}$ | $99.51_{\pm0.02}$ / $99.13_{\pm0.11}$ | $99.82_{\pm0.02}$ / $99.89_{\pm0.10}$ | $94.69_{\pm0.04}$ / $82.23_{\pm1.17}$ |
| **CUB-Incomplete** | Vanilla CBM | $56.46_{\pm0.48}$ / $5.42_{\pm0.60}$ | $61.85_{\pm0.57}$ / $14.58_{\pm1.14}$ | $65.42_{\pm0.51}$ / $23.43_{\pm1.40}$ | $72.59_{\pm0.68}$ / $49.52_{\pm1.36}$ | $76.35_{\pm0.85}$ / $64.41_{\pm0.89}$ | $79.58_{\pm1.31}$ / $79.58_{\pm1.31}$ | $69.17_{\pm0.68}$ / $39.42_{\pm1.03}$ |
| | Hybrid CBM | $72.13_{\pm0.57}$ / $4.81_{\pm0.39}$ | $76.31_{\pm0.79}$ / $7.34_{\pm0.30}$ | $78.53_{\pm0.64}$ / $8.99_{\pm0.47}$ | $83.26_{\pm0.52}$ / $14.81_{\pm0.46}$ | $85.43_{\pm0.45}$ / $18.93_{\pm0.55}$ | $87.54_{\pm0.37}$ / $24.33_{\pm0.42}$ | $80.91_{\pm0.55}$ / $13.11_{\pm0.43}$ |
| | ProbCBM | $60.56_{\pm1.11}$ / $4.26_{\pm0.96}$ | $67.50_{\pm0.59}$ / $13.06_{\pm1.47}$ | $71.80_{\pm0.61}$ / $23.37_{\pm1.64}$ | $80.55_{\pm0.54}$ / $55.05_{\pm1.10}$ | $84.45_{\pm0.33}$ / $72.47_{\pm0.45}$ | $87.80_{\pm0.09}$ / $87.80_{\pm0.09}$ | $76.01_{\pm0.40}$ / $42.46_{\pm1.05}$ |
| | P-CBM | $48.88_{\pm10.66}$ / $2.76_{\pm0.35}$ | $37.68_{\pm8.40}$ / $5.65_{\pm0.37}$ | $34.49_{\pm8.29}$ / $8.18_{\pm1.12}$ | $30.03_{\pm8.61}$ / $15.63_{\pm3.98}$ | $29.39_{\pm9.46}$ / $21.26_{\pm6.66}$ | $29.29_{\pm11.29}$ / $29.15_{\pm10.90}$ | $33.58_{\pm8.98}$ / $13.55_{\pm3.52}$ |
| | Residual P-CBM | $70.69_{\pm0.21}$ / $4.82_{\pm2.01}$ | $72.26_{\pm0.75}$ / $7.09_{\pm0.93}$ | $73.28_{\pm1.20}$ / $8.71_{\pm0.12}$ | $75.07_{\pm2.32}$ / $12.28_{\pm2.63}$ | $76.01_{\pm2.94}$ / $15.21_{\pm5.03}$ | $77.18_{\pm3.73}$ / $18.37_{\pm7.83}$ | $74.25_{\pm1.83}$ / $11.10_{\pm2.05}$ |
| | Bayes Classifier | $0.52_{\pm0.00}$ / $0.52_{\pm0.00}$ | $7.82_{\pm0.44}$ / $7.82_{\pm0.44}$ | $18.65_{\pm1.61}$ / $18.65_{\pm1.61}$ | $55.53_{\pm1.95}$ / $55.53_{\pm1.95}$ | $73.92_{\pm1.18}$ / $73.92_{\pm1.18}$ | $87.56_{\pm0.33}$ / $87.56_{\pm0.33}$ | $40.21_{\pm1.01}$ / $40.21_{\pm1.01}$ |
| | CEM | $74.42_{\pm0.46}$ / $6.84_{\pm1.69}$ | $79.96_{\pm0.68}$ / $12.92_{\pm2.62}$ | $82.89_{\pm0.43}$ / $17.52_{\pm2.96}$ | $87.92_{\pm0.75}$ / $29.71_{\pm3.52}$ | $89.99_{\pm0.51}$ / $36.81_{\pm3.78}$ | $92.08_{\pm0.47}$ / $44.61_{\pm4.44}$ | $85.09_{\pm0.65}$ / $24.87_{\pm3.18}$ |
| | IntCEM | $72.61_{\pm0.21}$ / $7.98_{\pm0.32}$ | $81.35_{\pm0.19}$ / $20.12_{\pm1.15}$ | $85.02_{\pm0.28}$ / $29.82_{\pm1.65}$ | $91.58_{\pm0.30}$ / $54.21_{\pm3.51}$ | $94.02_{\pm0.28}$ / $66.47_{\pm3.35}$ | $96.08_{\pm0.34}$ / $78.03_{\pm3.15}$ | $87.69_{\pm0.25}$ / $43.34_{\pm2.23}$ |
| | MixCEM (ours) | $74.56_{\pm0.12}$ / $11.61_{\pm1.37}$ | $83.51_{\pm0.30}$ / $27.73_{\pm2.36}$ | $87.02_{\pm0.15}$ / $40.74_{\pm2.76}$ | $92.80_{\pm0.17}$ / $68.61_{\pm2.71}$ | $94.91_{\pm0.14}$ / $80.51_{\pm1.99}$ | $96.64_{\pm0.19}$ / $90.42_{\pm1.20}$ | $89.16_{\pm0.16}$ / $54.48_{\pm2.13}$ |
| **AwA2** | Vanilla CBM | $87.52_{\pm0.41}$ / $15.99_{\pm0.86}$ | $92.17_{\pm0.47}$ / $33.86_{\pm2.63}$ | $96.59_{\pm0.26}$ / $71.70_{\pm2.86}$ | $98.71_{\pm0.16}$ / $93.62_{\pm1.33}$ | $99.36_{\pm0.09}$ / $99.30_{\pm0.23}$ | $99.49_{\pm0.13}$ / $99.49_{\pm0.13}$ | $96.18_{\pm0.21}$ / $71.31_{\pm1.50}$ |
| | Hybrid CBM | $88.18_{\pm0.65}$ / $17.30_{\pm0.66}$ | $91.26_{\pm0.39}$ / $25.89_{\pm2.05}$ | $94.38_{\pm0.29}$ / $45.18_{\pm4.41}$ | $96.70_{\pm0.21}$ / $69.67_{\pm5.25}$ | $98.83_{\pm0.03}$ / $92.65_{\pm2.61}$ | $99.53_{\pm0.06}$ / $99.26_{\pm0.11}$ | $95.08_{\pm0.23}$ / $58.52_{\pm2.88}$ |
| | ProbCBM | $85.34_{\pm0.39}$ / $6.41_{\pm1.76}$ | $90.44_{\pm0.38}$ / $29.07_{\pm7.21}$ | $96.34_{\pm0.21}$ / $72.61_{\pm7.46}$ | $99.01_{\pm0.04}$ / $93.80_{\pm2.22}$ | $99.95_{\pm0.01}$ / $99.78_{\pm0.10}$ | $100.00_{\pm0.00}$ / $100.00_{\pm0.00}$ | $95.78_{\pm0.15}$ / $69.81_{\pm3.67}$ |
| | P-CBM | $90.31_{\pm0.12}$ / $19.78_{\pm1.67}$ | $95.07_{\pm0.42}$ / $47.41_{\pm1.69}$ | $97.67_{\pm0.08}$ / $77.87_{\pm0.34}$ | $98.62_{\pm0.13}$ / $91.69_{\pm0.23}$ | $99.08_{\pm0.37}$ / $97.76_{\pm0.21}$ | $99.37_{\pm0.89}$ / $99.37_{\pm0.89}$ | $97.17_{\pm0.13}$ / $75.35_{\pm0.42}$ |
| | Residual P-CBM | $90.60_{\pm0.14}$ / $19.49_{\pm1.84}$ | $94.73_{\pm0.23}$ / $42.24_{\pm2.06}$ | $97.43_{\pm0.07}$ / $71.72_{\pm1.12}$ | $98.55_{\pm0.09}$ / $88.35_{\pm0.83}$ | $99.21_{\pm0.29}$ / $96.69_{\pm0.64}$ | $99.73_{\pm0.22}$ / $99.39_{\pm0.21}$ | $97.14_{\pm0.06}$ / $72.06_{\pm1.09}$ |
| | Bayes Classifier | $1.18_{\pm1.01}$ / $1.18_{\pm1.01}$ | $12.98_{\pm0.80}$ / $12.98_{\pm0.80}$ | $55.74_{\pm1.19}$ / $55.74_{\pm1.19}$ | $89.23_{\pm0.63}$ / $89.23_{\pm0.63}$ | $99.60_{\pm0.15}$ / $99.60_{\pm0.15}$ | $100.00_{\pm0.00}$ / $100.00_{\pm0.00}$ | $61.50_{\pm0.40}$ / $61.50_{\pm0.40}$ |
| | CEM | $91.07_{\pm0.24}$ / $20.22_{\pm2.03}$ | $94.36_{\pm0.56}$ / $25.05_{\pm2.24}$ | $95.10_{\pm0.63}$ / $28.76_{\pm1.98}$ | $96.46_{\pm0.27}$ / $33.44_{\pm2.17}$ | $98.00_{\pm0.13}$ / $39.40_{\pm2.30}$ | $98.86_{\pm0.08}$ / $44.81_{\pm2.05}$ | $95.60_{\pm0.19}$ / $31.64_{\pm2.07}$ |
| | IntCEM | $89.52_{\pm0.92}$ / $16.06_{\pm1.41}$ | $94.36_{\pm0.56}$ / $25.05_{\pm2.24}$ | $97.31_{\pm0.28}$ / $36.43_{\pm3.14}$ | $98.54_{\pm0.22}$ / $46.16_{\pm3.66}$ | $99.39_{\pm0.07}$ / $57.34_{\pm3.45}$ | $99.64_{\pm0.07}$ / $64.68_{\pm3.24}$ | $96.97_{\pm0.29}$ / $41.26_{\pm2.99}$ |
| | MixCEM (ours) | $89.97_{\pm0.13}$ / $17.75_{\pm0.18}$ | $95.45_{\pm0.11}$ / $45.51_{\pm0.59}$ | $98.83_{\pm0.02}$ / $82.73_{\pm0.69}$ | $99.62_{\pm0.04}$ / $96.95_{\pm0.50}$ | $99.99_{\pm0.00}$ / $99.91_{\pm0.05}$ | $100.00_{\pm0.00}$ / $100.00_{\pm0.00}$ | $97.92_{\pm0.02}$ / $77.17_{\pm0.37}$ |
| **AwA2-Incomplete** | Vanilla CBM | $76.28_{\pm0.82}$ / $15.43_{\pm0.38}$ | $73.92_{\pm1.11}$ / $27.22_{\pm0.17}$ | $72.90_{\pm1.24}$ / $35.29_{\pm0.07}$ | $71.90_{\pm1.16}$ / $46.30_{\pm0.58}$ | $70.66_{\pm1.37}$ / $58.13_{\pm0.42}$ | $69.63_{\pm1.61}$ / $69.63_{\pm1.61}$ | $72.39_{\pm1.20}$ / $42.87_{\pm0.25}$ |
| | Hybrid CBM | $89.39_{\pm0.16}$ / $18.11_{\pm1.19}$ | $90.26_{\pm0.23}$ / $20.60_{\pm1.28}$ | $90.78_{\pm0.17}$ / $22.06_{\pm1.26}$ | $91.34_{\pm0.18}$ / $24.02_{\pm1.43}$ | $91.88_{\pm0.15}$ / $26.19_{\pm1.66}$ | $92.41_{\pm0.07}$ / $28.64_{\pm1.52}$ | $91.09_{\pm0.17}$ / $23.48_{\pm1.37}$ |
| | ProbCBM | $67.12_{\pm0.18}$ / $7.61_{\pm1.80}$ | $69.59_{\pm0.32}$ / $18.82_{\pm1.13}$ | $70.82_{\pm0.27}$ / $29.39_{\pm1.40}$ | $72.48_{\pm0.40}$ / $43.85_{\pm0.74}$ | $73.83_{\pm0.13}$ / $60.62_{\pm0.57}$ | $75.58_{\pm0.47}$ / $75.58_{\pm0.47}$ | $71.82_{\pm0.13}$ / $39.99_{\pm0.87}$ |
| | P-CBM | $75.77_{\pm0.44}$ / $14.20_{\pm0.46}$ | $70.27_{\pm0.34}$ / $26.85_{\pm1.04}$ | $70.48_{\pm0.35}$ / $36.16_{\pm0.82}$ | $71.22_{\pm0.35}$ / $48.10_{\pm0.47}$ | $72.32_{\pm0.16}$ / $61.28_{\pm0.91}$ | $73.76_{\pm0.77}$ / $73.76_{\pm0.77}$ | $71.69_{\pm0.20}$ / $44.27_{\pm0.64}$ |
| | Residual P-CBM | $89.56_{\pm0.17}$ / $18.37_{\pm0.98}$ | $92.69_{\pm0.15}$ / $33.31_{\pm1.87}$ | $93.99_{\pm0.18}$ / $42.48_{\pm2.36}$ | $95.09_{\pm0.06}$ / $52.96_{\pm3.00}$ | $96.06_{\pm0.14}$ / $64.50_{\pm2.53}$ | $96.95_{\pm0.27}$ / $76.00_{\pm1.94}$ | $94.32_{\pm0.13}$ / $49.07_{\pm2.13}$ |
| | Bayes Classifier | $1.19_{\pm0.61}$ / $1.19_{\pm0.61}$ | $14.17_{\pm0.83}$ / $14.17_{\pm0.83}$ | $30.08_{\pm0.83}$ / $30.08_{\pm0.83}$ | $47.82_{\pm1.05}$ / $47.82_{\pm1.05}$ | $63.64_{\pm0.42}$ / $63.64_{\pm0.42}$ | $76.35_{\pm0.35}$ / $76.35_{\pm0.35}$ | $39.56_{\pm0.50}$ / $39.56_{\pm0.50}$ |
| | CEM | $90.12_{\pm0.07}$ / $17.42_{\pm0.82}$ | $92.47_{\pm0.06}$ / $22.09_{\pm0.83}$ | $93.57_{\pm0.13}$ / $24.92_{\pm0.90}$ | $94.61_{\pm0.18}$ / $28.07_{\pm1.11}$ | $95.64_{\pm0.23}$ / $31.59_{\pm1.44}$ | $96.67_{\pm0.26}$ / $35.45_{\pm1.72}$ | $94.04_{\pm0.14}$ / $26.97_{\pm1.07}$ |
| | IntCEM | $88.65_{\pm0.38}$ / $19.98_{\pm0.58}$ | $92.04_{\pm0.38}$ / $26.32_{\pm0.93}$ | $93.45_{\pm0.31}$ / $29.79_{\pm1.11}$ | $94.79_{\pm0.26}$ / $33.38_{\pm1.49}$ | $95.86_{\pm0.20}$ / $37.64_{\pm2.09}$ | $96.80_{\pm0.13}$ / $42.13_{\pm2.51}$ | $93.90_{\pm0.27}$ / $32.03_{\pm1.41}$ |
| | MixCEM (ours) | $88.65_{\pm0.43}$ / $16.71_{\pm0.82}$ | $93.01_{\pm0.20}$ / $38.74_{\pm0.50}$ | $94.84_{\pm0.15}$ / $45.38_{\pm0.47}$ | $96.30_{\pm0.13}$ / $58.95_{\pm0.71}$ | $97.51_{\pm0.04}$ / $72.44_{\pm0.84}$ | $98.50_{\pm0.04}$ / $83.74_{\pm1.10}$ | $95.20_{\pm0.10}$ / $53.07_{\pm0.41}$ |
| **CIFAR10** | Vanilla CBM | $76.97_{\pm0.17}$ / $35.35_{\pm2.07}$ | $81.58_{\pm0.40}$ / $51.25_{\pm0.72}$ | $86.55_{\pm0.23}$ / $71.98_{\pm0.50}$ | $88.88_{\pm0.12}$ / $81.61_{\pm0.18}$ | $89.86_{\pm0.10}$ / $87.86_{\pm0.18}$ | $89.31_{\pm0.12}$ / $89.31_{\pm0.12}$ | $86.11_{\pm0.11}$ / $71.31_{\pm0.36}$ |
| | Hybrid CBM | $79.12_{\pm0.27}$ / $31.59_{\pm1.47}$ | $80.07_{\pm0.27}$ / $35.00_{\pm1.54}$ | $80.89_{\pm0.31}$ / $39.17_{\pm1.81}$ | $81.58_{\pm0.24}$ / $43.09_{\pm1.75}$ | $82.71_{\pm0.25}$ / $50.18_{\pm1.53}$ | $83.61_{\pm0.24}$ / $55.24_{\pm1.30}$ | $81.36_{\pm0.28}$ / $42.30_{\pm1.58}$ |
| | ProbCBM | $64.80_{\pm5.15}$ / $27.40_{\pm2.71}$ | $70.70_{\pm4.77}$ / $37.02_{\pm1.26}$ | $78.38_{\pm3.81}$ / $54.75_{\pm2.96}$ | $83.85_{\pm2.59}$ / $69.50_{\pm3.47}$ | $89.91_{\pm0.45}$ / $86.58_{\pm0.57}$ | $90.87_{\pm0.11}$ / $90.87_{\pm0.11}$ | $80.27_{\pm2.81}$ / $61.68_{\pm1.01}$ |
| | P-CBM | $79.86_{\pm0.05}$ / $33.21_{\pm0.27}$ | $83.49_{\pm0.13}$ / $49.73_{\pm0.51}$ | $84.68_{\pm0.13}$ / $66.57_{\pm0.53}$ | $83.21_{\pm0.40}$ / $71.77_{\pm0.60}$ | $78.22_{\pm0.69}$ / $74.24_{\pm0.63}$ | $74.40_{\pm0.78}$ / $74.40_{\pm0.78}$ | $81.46_{\pm0.32}$ / $63.64_{\pm0.47}$ |
| | Residual P-CBM | $79.90_{\pm0.10}$ / $33.91_{\pm0.08}$ | $83.48_{\pm0.09}$ / $49.47_{\pm0.73}$ | $84.33_{\pm0.27}$ / $65.41_{\pm1.13}$ | $83.09_{\pm0.74}$ / $70.97_{\pm0.65}$ | $78.91_{\pm0.48}$ / $74.02_{\pm0.47}$ | $75.03_{\pm0.48}$ / $74.38_{\pm0.41}$ | $81.58_{\pm0.36}$ / $63.16_{\pm0.65}$ |
| | Bayes Classifier | $10.00_{\pm0.00}$ / $10.00_{\pm0.00}$ | $63.88_{\pm0.54}$ / $63.88_{\pm0.54}$ | $81.93_{\pm0.27}$ / $81.93_{\pm0.27}$ | $86.33_{\pm0.29}$ / $86.33_{\pm0.29}$ | $89.98_{\pm0.23}$ / $89.98_{\pm0.23}$ | $91.32_{\pm0.15}$ / $91.32_{\pm0.15}$ | $76.54_{\pm0.27}$ / $76.54_{\pm0.27}$ |
| | CEM | $80.05_{\pm0.35}$ / $32.93_{\pm0.14}$ | $83.36_{\pm0.34}$ / $42.21_{\pm0.20}$ | $86.46_{\pm0.23}$ / $54.57_{\pm0.48}$ | $88.17_{\pm0.33}$ / $62.69_{\pm0.59}$ | $89.99_{\pm0.13}$ / $71.52_{\pm0.57}$ | $90.58_{\pm0.16}$ / $75.26_{\pm0.30}$ | $86.73_{\pm0.23}$ / $57.30_{\pm0.36}$ |
| | IntCEM | $78.48_{\pm0.68}$ / $32.41_{\pm3.83}$ | $85.09_{\pm0.05}$ / $48.52_{\pm3.64}$ | $89.09_{\pm0.19}$ / $65.40_{\pm2.65}$ | $90.57_{\pm0.24}$ / $73.12_{\pm2.40}$ | $92.02_{\pm0.05}$ / $80.21_{\pm1.78}$ | $92.50_{\pm0.14}$ / $82.91_{\pm1.39}$ | $88.69_{\pm0.04}$ / $65.43_{\pm2.35}$ |
| | MixCEM (ours) | $79.36_{\pm0.69}$ / $32.79_{\pm1.04}$ | $83.49_{\pm0.60}$ / $48.75_{\pm1.43}$ | $87.37_{\pm0.58}$ / $67.55_{\pm1.22}$ | $89.34_{\pm0.44}$ / $77.48_{\pm0.83}$ | $91.37_{\pm0.41}$ / $85.78_{\pm0.40}$ | $92.51_{\pm0.39}$ / $88.85_{\pm0.31}$ | $87.66_{\pm0.52}$ / $68.46_{\pm0.88}$ |
| **CelebA** | Vanilla CBM | $24.18_{\pm0.69}$ / $10.71_{\pm0.67}$ | $28.39_{\pm2.12}$ / $14.37_{\pm2.09}$ | $30.66_{\pm2.32}$ / $18.46_{\pm2.65}$ | $33.20_{\pm3.10}$ / $24.83_{\pm3.00}$ | $36.28_{\pm2.99}$ / $31.73_{\pm2.16}$ | $40.18_{\pm2.79}$ / $40.18_{\pm2.79}$ | $32.45_{\pm2.44}$ / $23.59_{\pm2.43}$ |
| | Hybrid CBM | $35.43_{\pm0.23}$ / $10.73_{\pm1.00}$ | $35.79_{\pm0.21}$ / $10.79_{\pm1.02}$ | $35.89_{\pm0.23}$ / $10.79_{\pm1.02}$ | $36.02_{\pm0.21}$ / $10.80_{\pm1.02}$ | $36.20_{\pm0.25}$ / $10.85_{\pm1.04}$ | $36.35_{\pm0.26}$ / $10.91_{\pm1.04}$ | $35.97_{\pm0.22}$ / $10.82_{\pm1.03}$ |
| | ProbCBM | $31.74_{\pm0.29}$ / $12.92_{\pm1.00}$ | $39.79_{\pm0.47}$ / $19.48_{\pm1.77}$ | $44.37_{\pm0.06}$ / $23.81_{\pm2.17}$ | $48.56_{\pm2.26}$ / $31.11_{\pm2.32}$ | $55.61_{\pm0.08}$ / $42.27_{\pm2.29}$ | $62.96_{\pm0.12}$ / $62.96_{\pm0.12}$ | $48.00_{\pm0.10}$ / $32.60_{\pm1.62}$ |
| | P-CBM | $17.18_{\pm2.47}$ / $6.76_{\pm1.56}$ | $14.03_{\pm1.55}$ / $7.36_{\pm1.16}$ | $16.07_{\pm2.16}$ / $9.60_{\pm2.15}$ | $18.56_{\pm2.26}$ / $13.46_{\pm3.62}$ | $23.02_{\pm2.86}$ / $19.19_{\pm3.13}$ | $34.18_{\pm3.85}$ / $34.18_{\pm3.85}$ | $19.99_{\pm2.22}$ / $14.92_{\pm2.29}$ |
| | Residual P-CBM | $15.43_{\pm1.91}$ / $3.37_{\pm0.82}$ | $16.17_{\pm1.76}$ / $5.71_{\pm1.53}$ | $18.21_{\pm1.71}$ / $7.17_{\pm2.13}$ | $20.23_{\pm1.71}$ / $9.15_{\pm2.44}$ | $23.25_{\pm1.69}$ / $11.71_{\pm2.28}$ | $27.86_{\pm1.83}$ / $14.04_{\pm2.20}$ | $20.18_{\pm1.70}$ / $8.65_{\pm1.84}$ |
| | Bayes Classifier | $0.42_{\pm0.30}$ / $0.42_{\pm0.30}$ | $13.04_{\pm1.21}$ / $13.04_{\pm1.21}$ | $24.84_{\pm1.42}$ / $24.84_{\pm1.42}$ | $39.43_{\pm0.44}$ / $39.43_{\pm0.44}$ | $47.79_{\pm1.29}$ / $47.79_{\pm1.29}$ | $54.79_{\pm2.14}$ / $54.79_{\pm2.14}$ | $30.38_{\pm0.72}$ / $30.38_{\pm0.72}$ |
| | CEM | $34.89_{\pm0.46}$ / $6.52_{\pm2.94}$ | $40.58_{\pm0.48}$ / $7.60_{\pm3.18}$ | $43.66_{\pm0.50}$ / $8.35_{\pm3.39}$ | $47.34_{\pm0.32}$ / $9.12_{\pm3.53}$ | $51.02_{\pm0.73}$ / $10.22_{\pm3.56}$ | $54.82_{\pm0.57}$ / $11.78_{\pm3.81}$ | $45.85_{\pm0.36}$ / $9.02_{\pm3.43}$ |
| | IntCEM | $36.93_{\pm1.07}$ / $9.51_{\pm1.34}$ | $45.33_{\pm0.84}$ / $14.33_{\pm1.74}$ | $50.08_{\pm0.88}$ / $17.07_{\pm2.30}$ | $56.10_{\pm1.04}$ / $20.81_{\pm3.17}$ | $62.17_{\pm1.60}$ / $25.89_{\pm4.13}$ | $68.84_{\pm1.73}$ / $33.16_{\pm5.54}$ | $53.87_{\pm1.17}$ / $20.52_{\pm2.98}$ |
| | MixCEM (ours) | $35.53_{\pm0.76}$ / $11.76_{\pm0.74}$ | $44.17_{\pm0.50}$ / $18.11_{\pm1.13}$ | $49.17_{\pm0.35}$ / $22.90_{\pm1.80}$ | $55.48_{\pm0.39}$ / $30.95_{\pm2.17}$ | $61.83_{\pm0.39}$ / $42.93_{\pm1.76}$ | $69.15_{\pm0.68}$ / $62.53_{\pm0.57}$ | $53.24_{\pm0.43}$ / $31.99_{\pm1.21}$ |

