# OpenReview forum: "Avoiding Leakage Poisoning: Concept Interventions Under Distribution Shifts"
_ICML.cc/2025/Conference — ICML 2025 poster_

### Official Review · Reviewer_iEkK · 2025-03-13

**Overall Recommendation:** 3

**Summary:**

This paper investigates the behavior of concept-based models (CMs), particularly under distribution shifts, and introduces a novel model called MixCEM to address a critical limitation termed *leakage poisoning*. The authors demonstrate that existing state-of-the-art CMs, which rely on bypass mechanisms (e.g., residual connections) to handle incomplete concept annotations, suffer from degraded performance when intervened on out-of-distribution (OOD) samples. Leakage poisoning arises when leaked feature information becomes OOD, rendering concept interventions ineffective. MixCEM mitigates this by dynamically mixing *global* (sample-agnostic) and *contextual* (input-dependent) concept embeddings, using an entropy-based gating mechanism to suppress harmful residual information for OOD inputs. Experiments across four datasets (CUB, AwA2, CelebA, CIFAR10) and their concept-incomplete variants show that MixCEM outperforms baselines in both in-distribution (ID) and OOD accuracy, maintains intervenability, and avoids leakage poisoning. Key contributions include the formalization of leakage poisoning, the introduction of MixCEM, and the empirical validation of its robustness to distribution shifts.

**Claims And Evidence:**

The paper derives MixCEM’s objective function as a maximum likelihood estimation (MLE) under a probabilistic graphical model (Appendix B). While the derivation aligns with the proposed architecture, the proof is not explicitly detailed, and assumptions (e.g., independence between global embeddings and inputs) are stated without rigorous justification. The bounded intervenability (BI) property is introduced as a desirable criterion but lacks formal theoretical guarantees. Further analysis of how the entropy-based gating ensures OOD robustness would strengthen the theoretical foundation.

**Essential References Not Discussed:**

No essential references are not discussed.

**Experimental Designs Or Analyses:**

The experiments are extensive, covering ID/OOD scenarios, concept-complete/incomplete tasks, and multiple noise levels. Key strengths include:

- Comparisons against strong baselines (CEMs, IntCEMs, Hybrid CBMs, ProbCBMs).
- Use of synthetic OOD shifts (e.g., salt-and-pepper noise) and real-world distribution shifts (TravelingBirds).
- Ablation studies on hyperparameters (Appendix I) and bottleneck visualizations (Figure 5).

However, some aspects warrant clarification:

- The OOD noise injection method (Appendix G) uses pixel-level corruption, which may not fully represent real-world shifts (e.g., semantic changes).
- The Bayes classifier approximation (Appendix D.4) relies on a masked MLP; its fidelity to the true Bayes optimal performance is not validated.
- The reported improvements for CelebA are modest (e.g., ~35% task accuracy), suggesting potential limitations in highly noisy or subjective concept settings.

**Methods And Evaluation Criteria:**

**Methods**: MixCEM’s design is sensible for addressing leakage poisoning. By decoupling global and contextual embeddings, the model dynamically adjusts reliance on leaked features, which aligns with the goal of balancing completeness-agnosticism and intervenability. The entropy-based gating mechanism is a novel and logical approach to suppress OOD residuals.
**Evaluation**: The benchmarks (CUB, AwA2, CelebA, CIFAR10) are standard in concept-based XAI, and the inclusion of concept-incomplete variants and synthetic/OOD shifts (e.g., TravelingBirds, salt-and-pepper noise) is appropriate. However, the evaluation focuses on synthetic noise and spurious correlations; testing on natural distribution shifts (e.g., domain adaptation datasets like PACS) would strengthen validity.

**Other Comments Or Suggestions:**

No other comments.

**Other Strengths And Weaknesses:**

**Weaknesses**:

- **Computational Overhead**: MixCEM’s residual dropout and Monte Carlo sampling (Appendix I.3) increase inference time, which is not quantified.
- **Hyperparameter Sensitivity**: While ablations show robustness, MixCEM requires tuning $\lambda_p$$p_{drop}$ , and $E_{cal}$, which may limit accessibility.
- **Limited Real-World Shifts**: Experiments rely on synthetic noise; testing on natural OOD data (e.g.,  ImageNet-A[1]) would bolster claims.

[1] Benchmarking Neural Network Robustness to Common Corruptions and Perturbations

**Questions For Authors:**

No further questions.

**Relation To Broader Scientific Literature:**

The work builds on concept-based XAI (CBMs, CEMs) and addresses gaps in handling distribution shifts—an understudied aspect in interpretable ML. It connects to broader literature on OOD generalization and intervention-aware models (e.g., IntCEMs). The leakage poisoning concept parallels issues in robust representation learning (e.g., disentanglement of spurious correlations). However, the discussion could better contextualize MixCEM’s contributions relative to recent advances in OOD detection and causal intervention frameworks.

**Theoretical Claims:**

I did not check the details of the theoretical proof.

---

> ### Author Rebuttal · Authors · 2025-03-31
>
> Thank you for taking the time to give us very insightful feedback. We are glad you found our architecture a “sensible” solution to solve the task at hand and its gating mechanism “novel”. Moreover, we are happy to read that you found our evaluation “extensive.” Below, we focus on addressing some of the concerns you raised in your review. If you have further questions or concerns, please let us know. Otherwise, we would sincerely appreciate it if you would consider updating your score accordingly.
>
> ### **(W1) Inference Overhead**
>
> In Appendix F, we mainly studied the training cost of MixCEM’s architectural components w.r.t. competing methods. However, we agree that this could also discuss inference costs.  We note that the MC sampling in MixCEM is a relatively cheap operation as it does not require rerunning the bottleneck generator.  We see this [$\underline{\textbf{here}}$](https://imgur.com/a/D2bEjg4) when we look at the inference wall-clock times of all baselines in CUB. Although there is a slight increase in inference times of MixCEMs w.r.t. CEMs (~4.2% slower), we argue this difference is not problematic considering MixCEM is designed to be deployed together with an expert that can intervene on it. In this setup, less than a millisecond of extra latency should not bear too heavy of a toll, as post-intervention accuracy is more important. Nevertheless, we will update Appendix F to include this discussion.
>
> ### **(W2) Hyperparameter Sensitivity**
>
> We agree that MixCEM has some hyperparameters that need fine-tuning. However, we show in Appendix I that MixCEM’s performance is very robust to a series of hyperparameters and, more importantly, **in Appendix I.6 we provide recommendations for selecting hyperparameters if one does not have the time to fine-tune the model**. There, we suggest focusing only on $\lambda_c$. This means that future work could easily use our model (whose implementation as a standalone layer, seen in our code submission, will be made public) by trying only a few values for $\lambda_c$ and fixing all other hyperparameters to our recommended default values. We hope this will encourage the ease of use and accessibility of our proposed methodology.
>
> ### **(W3) Evaluation of more complex distribution shifts**
>
> Thank you so much for bringing up this excellent point! **We have now performed new experiments that show that our results hold across several forms of realistic image noise/shifts (e.g., downscalings, affine transforms, etc.) as well as in different forms of noise like domain shifts (e.g., an MNIST-trained model deployed on real-world SVHN digits)**. Please refer to our reply to Q1 of Reviewer KTVL for these results.
>
> ### **Improvements in CelebA**
>
> As argued in our reply to “Concern 1” of Reviewer At2h (please refer to this reply for details), we are interested in improving task intervention accuracy in ID and, especially, OOD test sets rather than unintervened accuracy. We want to achieve this without significantly dropping unintervened accuracy. So, while we agree that the improvements of unintervened accuracy in CelebA are modest w.r.t. IntCEM and CEM, we highlight that, when all concepts are intervened on in the OOD CelebA test set, MixCEM’s task accuracy is 62.53%. **This is about 29% and 51% percentage points (!) above the accuracies of IntCEM and CEMs, respectively, when all concepts are intervened on those models too**.
>
> ### **Bayes Classifier**
>
> We used our masked MLP evaluation mostly because we needed a tractable way of testing a model that takes as an input **any** concept subset and predicts a downstream task from it. Given the limited compute and data we had for each task, it is intractable to calculate the true Bayes error optimally. Nevertheless, we point out that even if this baseline is not accurate and therefore underperforms w.r.t. the true Bayes Classifier, we still observe that several key concept-based models (e.g., CEMs, IntCEMs, and P-CBMs) significantly underperform against this approximation when intervened on in OOD test sets (our main claim in Section 5.2). This means they are unlikely to perform better than the true Bayes Classifier.
>
> We will clarify this important point in our updated manuscript by better motivating using a masked MLP approximation for the Bayes Classifier in Section 5.
>
> ### **Theoretical Guarantees for BI**
>
> Our main goal was to highlight a key design limitation in modern CBMs (that of interventions not working when inputs go OOD) that had not been pointed out in the 3+ years that residual and embedding concept-based architectures have existed. BI serves as a way to formalize a target that can help us study this design consideration. As such, we opted to provide a guideline for a research direction that may lead to achieving BI using a novel architectural gating mechanism. Nevertheless, we will make sure to point out that future work could explore formal guarantees for BI in MixCEM and similar models as we agree these are important.

---

### Official Review · Reviewer_NChs · 2025-03-14

**Overall Recommendation:** 3

**Summary:**

The paper proposes MixCEM (Mixture of Concept Embeddings Model), which uses an entropy-based gating mechanism to control the leakage of information from the feature extractor. MixCEM is designed to dynamically adjust the influence of residual (leaked) embeddings so that they are beneficial for in-distribution samples while being suppressed for OOD samples. The authors back their claims with a thorough experimental evaluation on multiple datasets using diverse baseline CBM architectures.

**Claims And Evidence:**

Yes

**Essential References Not Discussed:**

All references discussed.

**Experimental Designs Or Analyses:**

Yes

**Methods And Evaluation Criteria:**

Yes

**Other Comments Or Suggestions:**

Refer Weakness.

**Other Strengths And Weaknesses:**

Strengths:
1. The identification of leakage poisoning is novel and highlights a critical, previously overlooked issue in the design of CBMs.
2. The proposed solution, MixCEM, introduces an innovative gating mechanism that adapts to the uncertainty in concept predictions, thereby improving both ID and OOD performance.
3. Paper is clear and easy to read.

Weakness:
1. Limited Novelty: The architecture clearly builds on top of CEM [1] with the presence/absence concept embeddings ($\hat{c}$) contextualized with residual embeddings ($r$) that correspond to residual embeddings and global embeddings ($c$) which are learnable. The entropy-based gating mechanism controls ID and OOD samples' residual embeddings. This introduces more parameters and alternative reasoning pathways for OOD samples - which is overall limited in novelty.
2. Comparisons to other CBM+OOD detectors - Approaches like GlanceNets [2] also fix leakage using a OOD detection mechanism. Hope the authors can provide a comparison to such approaches.
3. Results: In most of the datasets, CEMs actually perform better than MixCEM. If this is the case, can a OOD detector as used in [2] be useful in discarding them to improve performance?

[1] Concept Embedding Models, NeurIPS 22
[2] GlanceNets, NeurIPS 22

**Questions For Authors:**

Refer Weakness

**Relation To Broader Scientific Literature:**

It relates well with current state of research.

**Theoretical Claims:**

Yes

---

> ### Author Rebuttal · Authors · 2025-03-31
>
> Thank you so much for your insightful review! Your comments, particularly those regarding OOD detectors, have helped us improve our manuscript.
>
> We are glad you found our identification of leakage poisoning novel and telling of a “previously overlooked issue in the design of CBMs.” Moreover, we are glad you appreciate the novelty behind our entropy-based gating mechanism and that you believe this paper was “clear and easy to read.”
>
> Below, we reply to the concerns raised as part of your review. If you have further questions or concerns, please let us know. Otherwise, we would appreciate it if you would consider updating your score.
>
> ### **(W1) Limited Novelty of Architecture**
>
> As discussed in our “Summary of Contributions” subsection, we see the MixCEM architecture as just one of our contributions. In particular, we believe the study of concept interventions in OOD setups and the identification of “leakage poisoning” (something not previously identified in the 3+ years these architectures have existed) are equally important contributions to this work.
>
> We see the simplicity of MixCEM’s additions vis-a-vis CEMs as a “feature” rather than a “bug”. This work shows that a simple yet well-motivated modification to the CEM architecture can lead to models that are not only significantly better than CEMs at receiving interventions in OOD samples but **they are better than CEMs at receiving interventions even for in-distribution test sets**! This was achieved by introducing a simple yet novel entropy gating mechanism (as pointed out by this review). Considering all of this, we believe this work provides a series of novel contributions and describes an important yet previously unknown design consideration for concept-based models.
>
> ### **(W2) CBM + OOD Detector Approaches**
>
> Regarding GlanceNets, we opted against including them for two reasons: First, they are not completeness-agnostic models, as their label prediction is based only on the concept-aligned latent factors. Second, although GlanceNets can detect leakage via their Open Set Recognition module, once this leakage is detected, the architecture does not provide a solution that allows operations like interventions to work in that instance. For example, in the leakage experiments (Section 5.2) of the [GlanceNet paper](https://arxiv.org/abs/2205.15612), the authors “implement rejection by predicting a random label“, meaning that if the OOD detector is triggered (i.e., a sample is “rejected”), GlanceNet simply outputs a random label.
>
>
> Nevertheless, we believe your point about using a CBM and an OOD detector is a good suggestion and something one may want to try. As seen in the GlanceNet example above, however, even if we have a perfect OOD detector, we need to know how to act given the knowledge that an input might be OOD. In architectures like CEMs, one could, in theory, act on this knowledge by intervening using average “global” concept-level embeddings rather than the dynamic sample-dependent embeddings CEMs use for interventions (as this will “destroy” all leakage). We attempted this by **running new experiments on our CUB and AwA2 tasks, and we report the results of our strongest baselines, for visual clarity, [in this plot](https://imgur.com/a/k3nAzME)**. Here, we assume we have an “oracle” OOD detector that is always right for CEM, and use the mean active/inactive concept embedding computed in the training set to perform an intervention whenever the OOD oracle is triggered. Our results suggest that **even when we have a perfect OOD detector, interventions on CEMs are significantly worse than those in MixCEMs when samples are OOD**. More importantly, this strategy can lead to even worse intervention performance than using CEM’s original dynamic embedding when intervening. These results strongly suggest there is a significant benefit in jointly learning global and residual embeddings for interventions, as MixCEM does, and in including an OOD-detector-like gating mechanism as part of the inference path (as done by our entropy-based gates).
>
> Given the above, we will (1) update Section 5’s “Baselines” subsection to clarify why GlanceNets are not an ideal baseline for our evaluation, and (2) discuss in Section 5.2 why OOD detectors, on their own, may not work unless architectural changes are made, with the results of the new experiments included in a new Appendix.
>
>
> ### **(W3) MixCEM vs CEM**
>
> Thank you for bringing up this point! We would appreciate it if you could refer to our reply to a similar question by reviewer At2h (our reply to “*Concern 1*”). In summary, we argue that MixCEM’s performance is significantly better than CEM’s when **intervened on** for ID and OOD datasets (**up to 55% percentage points in task accuracy!**). We hope that this discussion, together with the discussion above on why it is not enough to have an OOD detector with CEM to make interventions work for OOD samples, clarifies why MixCEMs bring significant improvements over CEMs.

---

### Official Review · Reviewer_KTVL · 2025-03-22

**Overall Recommendation:** 4

**Summary:**

The authors present the first study examining the effectiveness of concept interventions under distribution shifts in interpretable concept-based models introducing the concept of "leakage poisoning", a phenomenon that hinders models from accurately improving when intervened upon for out-of-distribution inputs. To address this challenge, they propose the Mixture of Concept Embeddings Model (MixCEM), a new model architecture built upon recent concept embeddings models (CEMs). The proposed architecture is designed to adaptively leverage leaked information missing from its concepts only when this information is in-distribution.  Through a comprehensive evaluation covering both concept-complete and concept-incomplete tasks, the authors illustrate that MixCEM enhances accuracy for both in-distribution and out-of-distribution samples, regardless of the presence or absence of concept interventions. Moreover, it bridges the performance gap in state-of-the-art CEM models when handling out-of-distribution samples during concept interventions.


## update after rebuttal
I acknowledge the authors' response to my concerns, including the additional distribution shift experiment. Their clarifications show that the proposed methods effectively address performance gaps for out-of-distribution samples during concept interventions, enhancing explainability applications. Considering the contributions and the rebuttal response, my score is changed from "weak accept" to "accept", though something may escape my evaluation since I'm not an expert in concept-based models and their applications.

**Claims And Evidence:**

The authors assert that existing CEM models struggle to manage both concept incompleteness and test-time interventions when dealing with out-of-distribution inputs, whereas MixCEM remains unaffected by these challenges. To confirm this claim, they evaluate the proposed architecture alongside different concept-based models (including CEMs) across varying proportions of intervened concepts in both in-distribution and out-of-distribution settings.

**Essential References Not Discussed:**

The paper provides a well-structured overview of various categories of CMs, including CBMs, their extensions, and CEMs.

**Experimental Designs Or Analyses:**

The experimental design and analysis reported in the main paper are sound, though a more extensive evaluation across a wider range of out-of-distribution settings would further strengthen the study.

**Methods And Evaluation Criteria:**

The proposed evaluation criteria make sense for the problem, but a more comprehensive evaluation of different out-of-distribution settings would strengthen the paper’s impact. The analysis encompasses both concept-complete and concept-incomplete datasets, as well as two additional datasets, one synthetic and one real-world, where concept attributes are derived from either CLIP-based classification or subjective human annotations. However, the paper lacks a broader range of distribution shifts (which is the focus of the paper) beyond salt-and-pepper noise. Regarding the method’s comparison, the study evaluates various models, including Deep Neural Network (DNN), Concept Embedding Models (CEMs), and Concept Bottleneck Models (CBMs).

**Other Comments Or Suggestions:**

Adding a diagram illustrating how concept interventions work could enhance understanding.

**Other Strengths And Weaknesses:**

Strengths:
- The paper is well-written and well-structured, with clearly articulated contributions.
- The paper proposes a novel approach to handling distribution shifts within CMs maintaining competitive performances across different scenarios.

Weaknesses:
- The experimental comparison lacks the inclusion of label-free CBMs and energy-based CBMs mentioned in the "CBMExtensions" paragraph.
- The study does not extensively evaluate the method’s robustness across a broader range of distribution shifts beyond salt-and-pepper noise. The discussion on a different distribution shift (TravelingBirds) is very limited.

**Questions For Authors:**

1. How does your approach handle more complex and varied distribution shifts beyond those tested in the paper? Is it possible to test other types of out-of-distribution datasets or samples? For example by using different domains and contexts, like training on MNIST and testing on SVHN or using shared object classes from Imagenet and CIFAR-10 datasets.


2. Under what conditions might your approach fail, and how could these limitations be addressed? Acknowledging potential weaknesses and offering solutions would strengthen the paper's robustness claims.

**Relation To Broader Scientific Literature:**

The paper positions its contribution in the context of literature on interpretability and distribution shift in concept-based models. Specifically, the authors claim to be the first to examine the impact of concept interventions on out-of-distribution samples, introducing the term "leakage poisoning". They then propose a novel architecture, comparing it with existing concept-based models and highlighting how it advances prior work by directly addressing leakage poisoning within CMs.

**Theoretical Claims:**

The theoretical foundation builds on prior work in interpretability and concept-based models, so there are no specific theoretical proofs that require verification. The focus is primarily on methodology and experimental outcomes.

---

> ### Author Rebuttal · Authors · 2025-03-31
>
> Thank you for your insightful feedback! Your comments really helped us improve the quality of our manuscript. We are glad you found our work novel, “well-written”, and “well-structured”. Below, we answer your main concerns. If you have further questions, please let us know. Otherwise, we would sincerely appreciate it if you would consider updating your score given our replies below.
> ### **(Q1/W2) Evaluation of other distribution shifts**
>
> In our evaluation, we focused primarily on salt and pepper noise as it is a common real-world form of noise [[1](https://arxiv.org/abs/1903.12261), [2](https://link.springer.com/article/10.1007/s11042-016-3622-9)]. However, our methodology makes no assumptions on the type of distribution shift, as seen in our TravelingBirds experiments. To provide further evidence for this, **we carried out these new experiments**:
> 1. **Other Noise Forms**: We evaluated interventions on samples that were *downsampled*, *gaussian blurred*, and applied a random *affine transformations* (rescalings and rotations). These are widespread noise forms in real images. Our [results](https://imgur.com/a/8SPvTd2) on CUB-Incomplete and our AwA2 tasks suggest that MixCEMs have better OOD intervention task accuracies than our baselines across different noise forms. For instance, MixCEMs can have up to ~20% percentage points more in OOD task accuracy than CEMs and IntCEMs when all concepts are intervened on in samples downsampled to 25% of their size. These results suggest that MixCEMs are better at receiving interventions in practical scenarios.
> 2. **Domain/Context Shifts**: We followed your insightful suggestion of testing our baselines when a domain shift occurs. We trained our models on an addition task where 11 MNIST digits form each training sample, and the task is to predict whether all digits add to more than 25% of the maximum sum.  We provide the identity of five digits as training concepts (i.e., it is an incomplete task), and at test time, we swap MNIST digits for SVHN digits. Our [results](https://imgur.com/a/Duoto5p) show that MixCEMs achieve better ID and OOD intervention task AUC-ROC than our baselines, particularly for high intervention rates. For example,  when all concepts are intervened, MixCEM attained ~31, ~7, and ~3 more percentage points in OOD task AUC-ROC over CEM, IntCEM, and ProbCBM, respectively. In contrast, we found it very difficult to get CEMs to perform well in this incomplete task.
>
> We will incorporate the results of (1) in a new Appendix summarised in §5.2 (where we will better motivate our use of S&P noise). We will discuss the results of (2) in §5.3, where they will complement our TravelingBirds experiments by showcasing MixCEM’s utility on different distribution shifts.
> ### **(W1) Energy-based and Label-free CBMs**
>
> Our evaluation focuses on baselines that cover key directions in concept learning: we include methods that are embedding-based (CEM/ProbCBM), intervention-aware (IntCEM), scalable (Posthoc CBM), and “traditional” (CBM variants). Moreover we **include label-free annotation pipelines** via our CIFAR experiments. We believe our **ten** evaluation baselines and diverse training sets cover an overview of competing methods. Because of this, we decided not to include Energy-based CBMs as they were not designed with intervention performance in mind and are outperformed by approaches such as CEMs (see Figure 2 of the [ECBM paper](https://arxiv.org/abs/2401.14142)).
>
> Label-free CBMs were not explicitly included in our evaluation for three reasons: (1) **their main contribution, i.e., using LLMs and VLMs for concept extraction, was included as part of our evaluation in our CIFAR dataset**; (2) in datasets where we have concept annotations (e.g., CUB), it is difficult to fairly compare label-free CBMs and concept-supervised methods as we do not have ground-truth labels for label-free concepts; and (3) Label-free CBMs were not designed with interventions in mind. This is exemplified by the original label-free CBM paper, which did not evaluate concept interventions.
>
> We will update our Baselines section to justify these decisions.
> ### **(Q2) Failure modes of MixCEMs**
>
> Although we discuss some limitations in §6, we agree that this section could examine potential failure modes. We foresee at least two limitations: (1) when a concept goes OOD and the shift renders the concept incomprehensible for an expert, MixCEMs may fail to completely block leakage poisoning as one cannot intervene on such a concept. Hence, future work can explore mechanisms for blocking all unwanted leakage without knowing a concept’s label.
>
> Second, in incomplete tasks, intervened MixCEMs do not always recover the full ID performance in OOD inputs. Therefore, future work could explore how to better limit leakage, rather than entirely remove it, so that information about unprovided concepts can still be exploited after an intervention.
>
> We will include this discussion in our Limitations section.

---

### Official Review · Reviewer_At2h · 2025-03-26

**Overall Recommendation:** 3

**Summary:**

This paper proposes a novel concept-based model called mixCEM. The authors point out the problem of leakage poisoning, where as concept interventions become out of training distribution, the task prediction accuracies reduce. The authors propose mixCEM which uses residual embeddings for positive and negative concept labels. These embedding are then mixed to create the concept predictor which further predicts the task.

**Claims And Evidence:**

The authors claim the proposed model mitigates the leakage posing problem. However, the theoretical argument towards this is not clear to me. Also, the experimental results in table 1 and figure 4 show that performance of CEM is comparable to mixCEM. The performance of mixCEM is sometimes better for OOD samples. However, I don't see a clear pattern.

**Essential References Not Discussed:**

None that I know of.

**Experimental Designs Or Analyses:**

The experiments look sound to me.

**Methods And Evaluation Criteria:**

I am not completely sure, but looks like the evaluation method is standard.

**Other Comments Or Suggestions:**

The writing is good.

**Other Strengths And Weaknesses:**

None.

**Questions For Authors:**

Can you explain crisply why mixing of residual embeddings are helpful?

**Relation To Broader Scientific Literature:**

The authors cite and connect with the literature in the area.

**Theoretical Claims:**

There are no theoretical claims.

---

> ### Author Rebuttal · Authors · 2025-03-31
>
> Thank you so much for taking the time to go over our work and provide feedback. Your comments have helped us identify areas where we can improve our manuscript.
>
> We are glad you found our paper “well-written” and its evaluation sound. Below, we reply to your feedback. If you have further questions or concerns, please let us know. Otherwise, we would sincerely appreciate it if you would consider updating your score after considering our replies.
>
> ### **(Concern 1) CEM vs MixCEM’s performance (what is the trend?)**
>
> As we point out in Section 5.1, “we do not expect MixCEM to be the best-performing baseline in terms of its in-distribution (ID) task and concept fidelity.”  This means that, in some datasets, CEM’s **unintervened performance on ID samples** is slightly higher than that of MixCEMs (at most 1.5% percentage points in task accuracy and 2% in concept accuracy). However, this slight drop in MixCEM’s unintervened performance is **statistically significant in only 3/6 tasks** (as determined by a paired t-test in Table 1). As we argue below, we see this as a reasonable trade-off considering MixCEM’s improvements in intervenability.
>
> Specifically, our work aims to improve OOD intervenability while remaining competitive against strong baselines such as CEMs in ID cases. Our results demonstrate that this is the case: across **all** datasets, MixCEMs achieve (statistically) significantly better accuracies than CEMs when intervened on, both in ID and OOD test sets. For example, in CUB alone, when one intervenes on 20% of the concepts on the OOD test set, MixCEM achieves a task accuracy of 62.24 ± 3.43% vs CEM’s 32.58 ± 3.12% (**an absolute improvement of nearly 30% points!**). Even in the ID test set, at an intervention rate of 20%, MixCEMs attain more than 3% of absolute gains w.r.t. CEMs. This gap is even wider for other datasets: in AwA2, MixCEM **has an OOD task accuracy ~55% higher (in percentage points!) than CEM’s** when all concepts are intervened on both models. Similar results can be seen across all datasets (see Figure 4 or Table 5, where we also provide statistical significance tests). We believe these improvements are worth a potential hit of 1-2% in unintervened accuracy if one expects the model to be intervened on at test time, our deployment setup of interest.
>
> Hence, the **observed trend** is the following: **the area under MixCEM’s intervention curves is significantly higher than CEM’s across all tasks, both in ID and OOD test sets (i.e., MixCEM is much more “intervenable” while remaining competitive without any interventions)**. You can see this by noticing that the red-dashed line (MixCEM) in Figure 4 **is always above** the green-triangle line (CEMs) when we perform one or more interventions.
>
> To clarify our results for future readers, we will update Section 5 and the caption of Figure 4 to explicitly make these points.
>
>
> ### **(Concern 2) Argument for why MixCEM mitigates leakage poisoning**
>
> Intuitively, MixCEM’s entropy gates work as OOD detectors that only let the residual component into an embedding when the sample is ID. Therefore, the embeddings used when intervening on OOD samples will have no residual components, meaning they will be “global” and not sample-specific embeddings. If no sample-specific information is allowed in the label predictor after an intervention, then leakage can't exist. This means that **by blocking leakage/residual information when a sample goes OOD, MixCEMs mitigates leakage poisoning**. Further theoretical motivation is provided in Appendix B.
>
> ### **(Q1) Can you explain crisply why mixing of residual embeddings is helpful?**
>
> Crisply put, we do mixing at two levels: (1) we construct final concept embeddings $\mathbf{c}_i$ by mixing *"positive" and "negative" semantic embeddings* $\mathbf{c}_i^{(+)}$ and $\mathbf{c}_i^{(-)}$, and (2) we construct each semantic embedding itself  $\mathbf{c}_i^{(+/-)}$ by mixing a *global embedding* $\bar{\mathbf{c}}_i^{(+/-)}$ and a corresponding *residual embedding* $r_i^{(+/-)}(\mathbf{h})$.
>
> These two levels of mixing allow one to (1) perform interventions by simply hardcoding a concept’s final embedding at test-time to $\mathbf{c}_i^{(+)}$, if the expert believes $c_i = 1$, and to $\mathbf{c}_i^{(-)}$ otherwise, and (2) avoid leakage poisoning by dropping residual components of each embedding when the sample is OOD (therefore completely blocking any leakage from influencing the downstream predictor). Notice that by allowing the residual compontent of each semantic embedding to be partially dropped (as it is mixed using a continuous value), this architecture allows MixCEM to use these residual embeddings to communicate useful information to the label predictor **both** for ID and OOD samples (leading to models that are completeness agnostic).

---

### Decision · Program_Chairs · 2025-05-01

**Decision:**

Accept (poster)

**Comment:**

This paper currently has three Weak Accept and one Accept recommendation.

Among the identified strengths, the paper is generally evaluated as well-written, easy to read, and structured. Some reviewers consider the contributions well-articulated, and acknowledged how this work successfully identifies some gaps and potential issues in how Concept-Based Models (CBM) interventions respond to out-of-distribution inputs, proposing an effective method based on the identification of  “leakage poisoning”, which is benchmarked against baselines and using both synthetic and real-world OOD shifts.
 In the first reviews, some reviewers raised concerns about computational overhead, the need for more experiments with a broader range of distribution shifts or real-world OOD shifts (e.g., ImageNet-A), the limited novelty of the proposed architecture, and comparisons to other CBM+OOD detectors.

In their rebuttal, the authors provide new experiments covering more distribution shifts beyond salt-and-pepper, more insights on CBM+OOD detectors, and clarifications on inference overhead and the main contributions and novelty.

Considering the reviewers’ scores and the rebuttal and evaluating strengths and weaknesses of the paper, the AC agrees on the identified strengths and recognizes the novelty in the identification of “leakage poisoning”; therefore, they recommend the acceptance of this work.